# Investigation of the Dynamic Pure-Mode-II Fracture Initiation and Propagation of Rock during Four-Point Bending Test Using Hybrid Finite–Discrete Element Method

Yushan Song [1] , Yuqing Fan [2], Huaming An [1,*] , Hongyuan Liu [3] and Shunchuan Wu [4,5]

1 Faulty of Public Security and Emergency Management, Kunming University of Science and Technology, Kunming 650093, China
2 School of Mining Engineering, Guizhou University of Engineering Science, Bijie 551700, China
3 College of Science and Engineering, University of Tasmania, Hobart, TAS 7001, Australia
4 Faculty of Land Resources Engineering, Kunming University of Science and Technology, Kunming 650093, China
5 School of Civil and Resource Engineering, University of Science and Technology Beijing, Beijing 100083, China
* Correspondence: huaming.an@kust.edu.cn; Tel.: +86-136-9912-4758

**Abstract:** A hybrid finite–discrete element method (FDEM) is proposed to investigate dynamic pure-mode-II fracture behaviors. The transition of continuum to discontinuum was applied to the FDEM through the use of three fracture modes, so that the whole fracture process could be modeled naturally. The FDEM was then employed to model the dynamic pure-mode-II fracture behavior of rock during a four-point bending test with a prefabricated notch. The results showed that the fracture initiated from the tip of the prefabricated notch under a relatively lower loading rate, i.e., 1 m/s and 5 m/s. However, when the loading rate reached higher levels, i.e., 10 m/s and 50 m/s, the prefabricated notch played a small role in the fracture patterns. Under these conditions, the fracture initiated from the center of the beam bottom or the stress concentration vicinity, instead of the tip of the prefabricated notch. Regardless of the loading rate, the obtained force-loading displacement curves showed a typical brittle material failure process. Additionally, by incorporating the empirical correlation between the static and dynamic strengths obtained from the dynamic rock fracture tests, the hybrid finite–discrete element method could effectively reflect the impact of the loading rate on the strength of the rock. To conclude, the hybrid finite–discrete element method is an effective instrument to investigate the fracture initiation and propagation of rock, since it can both naturally simulate the process of rock fracture and capture the effect of the loading rate on the rock behaviors.

**Keywords:** rock mechanics; dynamic fracture; crack propagation; pure mode II; four-point bending test; FDEM

## 1. Introduction

With the need for a thorough understanding of rock mass failure processes in both fundamental research and rock engineering designs, rock fracture mechanics has emerged as a promising offshoot of rock mechanics and fracture mechanics in recent years [1,2]. When a rock mass is subjected to dynamic loading conditions, the original fractures in the rock play a major role in the failure and collapse of the geostructures. Thus, a number of studies have been conducted to investigate the fracture mechanism of a rock mass under dynamic loading conditions [3–5]. Furthermore, the fracture processes initiated from a prefabricated notch or crack have been studied by numerical tools, i.e., RFPA [6] and FDEM [7], to obtain the influence of the existing crack on the rock failure behavior. In general, for simplified 2D problems, there are three main types of failure modes involved in the failure of rocks: pure mode I, pure mode II, and mixed mode I–II. Tensile failure and shear failure are examples of pure-mode-I and -II fractures, respectively, whereby the rock

exclusively fails under tensile stress or shear stress. The failure of a rock that occurs under both tensile and shear stress is referred to as mixed mode I–II. A number of researchers have studied these three failure modes through various experiments. Andrea et al. (2020) presented a pseudo-compact tension method to measure mode-I fracture toughness under pure tension conditions [8]. Liu et al. (2004) investigated the mode-II fracture propagation of brittle rock through both the shear box test and numerical simulation, and it was concluded that the heterogeneity and confinement played a key role in the formation and characteristics of the shear fracture [9]. Thus, the shear box test under confined conditions is an effective method to measure mode-II fracture toughness [9]. Applying the extended finite element approach, Xie et al. (2017) studied mixed-mode fracture behavior using a semi-circular bent specimen, and they thoroughly analyzed the crack initiation angle, crack propagation trajectory, and the start of the fracture [10]. However, in rock engineering, the shear fracture mode, i.e., pure mode II, can significantly cause larger failures and crack expansion in rock structures. In various rock engineering scenarios, such as the building of dams, underground structures, and tunnels, it is vital to evaluate the rock shear strength for the stability of geostructures [11]. In addition, in most rock bridges, failure between two adjacent discontinuities is caused by pure-mode-II brittle fractures [12]. Thus, it is imperative to study the fracture and fragmentation of geomaterials under pure-mode-II fracture, considering not only the stability of geostructures but also their destruction in terms of extracting valuable natural resources in the mining industry and geostructure demolition for urban reconstruction.

In the last few decades, numerous laboratory test methods have been implemented to research geomaterial fracture mechanisms, including the Brazilian tensile strength test, the Brazilian disc test with notches, the three- and four-point bending tests, and the uniaxial compressive strength test [13–15]. These test methods offer a greater comprehension of the characteristics of rocks, such as their strength and toughness. For further investigation of rock fracture characteristics, the numerical method has provided another way to understand the fracture process [16]. On the basis of the numerical model hypothesis, there are three common categories of numerical method: continuum-based methods, discontinuum-based methods, and hybrid or combination continuum–discontinuum-based methods [17–19]. Currently, many continuum methods, e.g., the finite element method (FEM) [20], the finite difference method (FDM) [21], the boundary element method (BEM) [22], the scaled boundary finite element method (SBFEM) [23], and the extended finite element method (XFEM) [24], have been employed to simulate geomaterial behaviors under different loading rates. The numerical models based on the discontinuum method consider complexes made up of discrete parts that are held together by cementation or cohesive forces [25]. The discontinuum methods have been employed to model collisions between discrete elements. The distinct element method (DEM) [19,26], the lattice model (LM) method [27], and molecular dynamics (MD) [28] are the most widely used discontinuum methods. The two abovementioned types of method have their advantages and disadvantages. The continuum method can effectively model the stress distribution, fracture initiation, and propagation before the post-failure of the geomaterials under loading, while the fracture interaction of the generated fragmentation and even the muck-piling after the post-failure can be accurately modeled using the discontinuum approach. Thus, the complete rock fracture and fragmentation process can be more accurately modeled using the hybrid continuum–discontinuum methods than the individual continuum or discontinuum methods. As a result, several multiscale coupling methods and hybrid approaches have been proposed and rapidly developed [18], such as the discrete element method/boundary element method (DEM-BEM), the discrete element method/finite element method (DEM-FEM), and the hybrid boundary element method/finite element method (BEM-FEM) [29].

The hybrid finite–discrete element approach (FDEM) is presented in this study to investigate the process of rock fracture under dynamic loading conditions, because the proposed method incorporates the advantages of both continuum-based methods and discontinuum-based methods. On the one hand, the FDEM can demonstrate the stress

distribution and describe elastic deformations, functions that are performed well by continuum methods. On the other hand, as with other discontinuum methods, the FDEM is an effective tool for dealing with interactions and collisions during the process of rock fracture and fragmentation. Due to these special advantages, the proposed method, which was initially established by Munjiza [30], has been applied not only to modeling rock mechanics experiments, such the Brazilian tensile strength test [7], the three- and four-point bending tests [16], and the uniaxial compression test [31], but also to handling geological engineering problems such as blasting [32–34], tunneling [35], and landslides [36]. For most geomaterials, e.g., rock and concrete, the compressive strength is 8 to 12 times greater than the tensile strength. Thus, geomaterials are mostly placed under compressive stress conditions in engineering applications. As mentioned before, the fractures in geomaterials include mode-I fractures, mode-II fractures, and mixed mode I–II fractures. In most geoengineering projects, the construction materials, e.g., rock and concrete, undergo compressive stress due to the fact that the compressive strength of rock-like materials is much higher than the tensile strength. Under compressive stress conditions, geomaterials mainly experience pure-mode-II fractures. Although we previously studied the rock failure pattern in a four-point bending (4 PB) test under a loading rate of 1 m/s [16], after a deeper exploration, we found that the loading rate played a significant role in the rock fracture behavior. Therefore, in this study, a 4 PB test's pure-model-II fracture process under different loading conditions is modeled using a numerical tool developed by the authors [37]. In addition, the empirical correlation between the static and dynamic strength is implanted into the FDEM in order to truly reflect the effect of various loading rates on rock failure behavior.

The organization of this paper is as follows. Section 2 presents the core concepts of the hybrid finite–discrete element method, with a particular focus on the transition from continuum to discontinuum through rock fracture, which is considered to be the key component of the method. Section 3 describes the numerical models for the four-point bending test and the application of various loading rates to the rock samples. In this section, the pure-mode-II fracture process is analyzed and the fracture toughness is obtained. In Section 4, the effect of the loading rate is discussed. Finally, the conclusions from this study are drawn. In the process of modeling a 4 PB test, a numerical model needs to be established, and all the rock parameters should be set up in advance. Next, different loading rates should be set on the two upper loading rolls. Through calculating the energy release rate of the mode-II fracture, which can be defined based on the empirical correlations, the sliding displacement of the crack mouth can be determined and the transition from continuum to discontinuum can be achieved. Finally, the rock sample's pure-mode-II fracture patterns under various loading rates can be illustrated visually. It is worth noting that the contact detection implanted in the FDEM is employed to reflect the interaction between the elements in order to save modeling time.

The FDEM is proposed by the authors for modeling conventional rock mechanical tests, e.g., the Brazilian dis test [7] and the uniaxial compression test [31], and for implementation in many engineering problems, e.g., tunnel excavation [33] and slope failure analysis [36]. Previous studies have focused on the macro-phenomena of rock mechanisms, while this study tries to illustrate the micro-phenomenon of the rock fracture process. This paper aims to demonstrate the capabilities of the FDEM in modeling the dynamic behavior throughout the course of a pure-mode-II fracture and the combined method's capacity to accurately capture how the loading rate affects dynamic fracture toughness. To successfully model the pure-mode-II fracture process, the transition from continuum to discontinuum, the three fracture modes, and the effect of the loading rate are considered.

## 2. Hybrid Finite–Discrete Element Method

The hybrid finite–discrete element method (FDEM) was initially developed by Munjiza (2004) to study the tensile fracture of concrete [30]. He then created the open-source Y2D/Y3D code for the FDEM application. Subsequently, many researchers have tried to expand the open-source code in order to advance the hybrid finite–discrete element

technique, such as with Y-GEO [38], IRAZU [39], Solidity [40], MUNROU [41], and Y-Flow [42]. A solitary discrete element or several discrete bodies with standard dimensions and shapes may be contained in the FDEM model. Each of these discrete bodies is then symbolized by a solitary discrete element [2]. The contact law is used to govern the interaction between discrete bodies. In this process, these discrete complexes are then discretized into finite elements in order to simulate the fracture and fragmentation of rock-like materials [2,36]. The main components of the FDEM include contact detection, the interaction of each body, the transition from continuum to discontinuum, deformability, a temporal integration scheme, and computational fluid dynamics [2,30,43–46]. The critical component, which is described in detail below, is the transition from a continuum to a discontinuum through fracture and fragmentation.

One or more discrete parts may be present in the hybrid finite–discrete element simulation, and these discrete parts are then discretized into many finite elements. Through the separation of the finite elements in the discrete bodies, the process of transition from continuum to discontinuum can be achieved. In order to model the mode-I fracture of concrete, Munjiza et al. (2004) presented the combined single and smeared crack model [30]. This combined single and smeared crack model was expanded by the authors of the present study to simulate various types of fractures. Figure 1 shows a standard stress–strain curve for a brittle material under loading, which is divided into the stress-hardening part and the stress-softening part [2]. In Figure 1, $\varepsilon_u$ is the ultimate strain, $\varepsilon_p$ is the strain at peak stress, and $f_p$ is the peak stress. A typical elastoplastic constitutive law is used to demonstrate the stress-hardening stage in the hybrid finite–discrete element technique. The stress-softening part is illustrated through three fracture models governed by the fracture energy release rates. The area under the stress-displacement curve indicates the energy release rate. The crack will spread if Equation (1) is met.

$$G_f = \int_{\varepsilon_p}^{\varepsilon_u} f(\varepsilon) \mathrm{d}\varepsilon \times l_c = 2\gamma \tag{1}$$

In this equation, $\varepsilon_p$ is the strain at peak stress, $\varepsilon_u$ is the ultimate strain, $\gamma$ is the surface energy, $G_f$ is the energy release rate, and $l_c$ is the characteristic length.

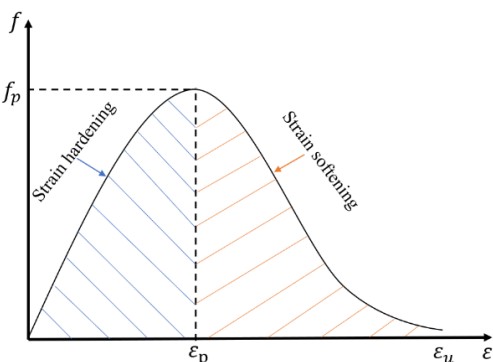

**Figure 1.** Standard geomaterial stress–strain curve under loading (after Liu et al. (2013) [2]).

Figure 2 illustrates the hybrid finite–discrete element model under different stress conditions [47]. The models are discretized by three-node triangle finite elements, which can fracture from the element edges to achieve the transition from continuum to discontinuum. As illustrated in Figure 2a, the finite elements are bonded using four-node cohesive elements, which can distort in normal and tangential directions to model the fracture onset and spread process. The stress used to bond finite elements is called bonding stress, which can be divided into normal and tangential directions. Equation (2) can be used to express how the finite elements are separated in the two directions.

$$\delta = \delta_n \boldsymbol{n} + \delta_s \boldsymbol{t} \tag{2}$$

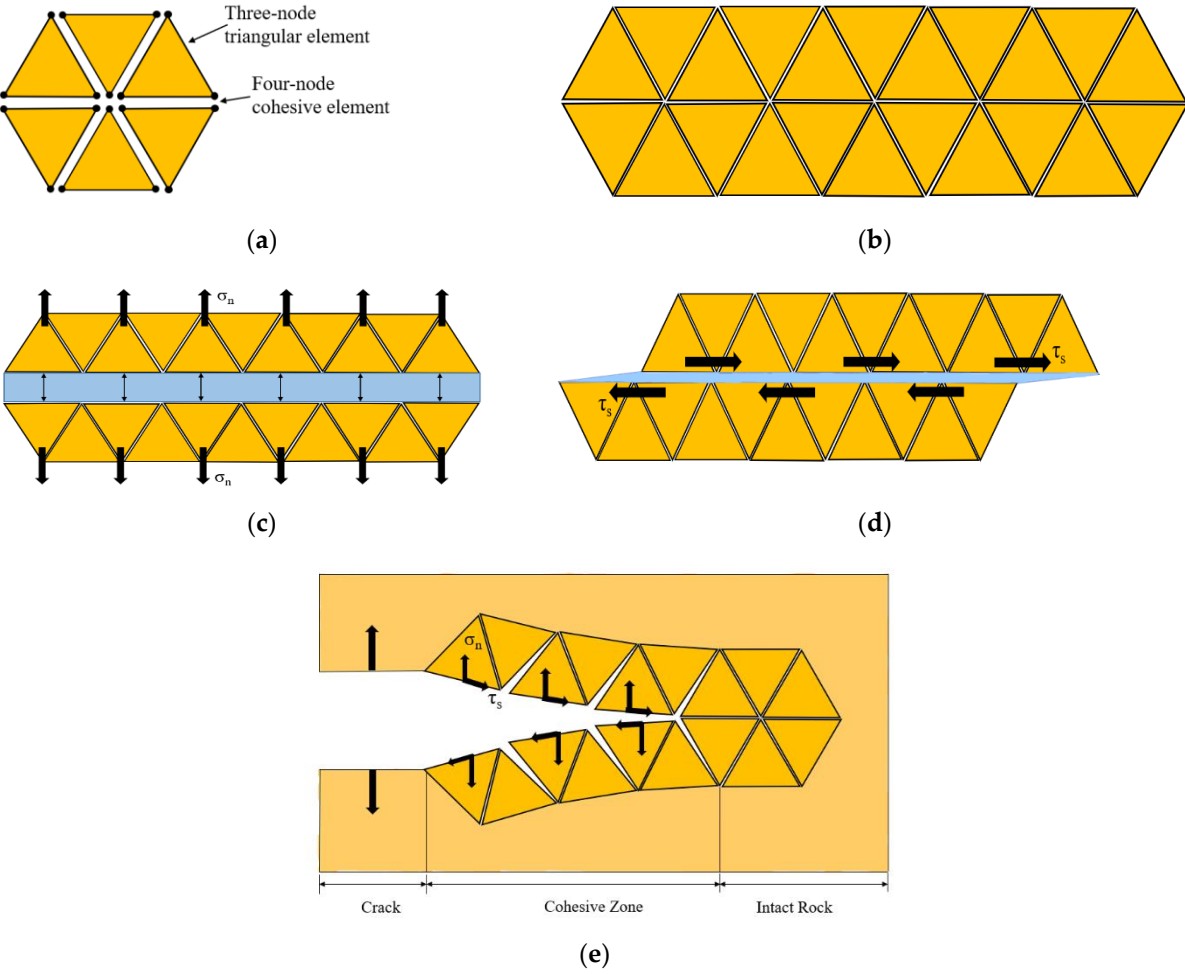

**Figure 2.** Conceptual FDEM and schematics under different stress conditions (after An et al. (2021) [47]). (**a**) Node distribution; (**b**) no stress; (**c**) under tension conditions; (**d**) under shear conditions; (**e**) under mixed tension and shear conditions.

In Equation (2), **n** represents the unit vectors in the normal direction; **t** represents the unit vectors in the tangential direction; $\delta$ is the separation of the adjacent finite elements; and $\delta_n$ and $\delta_s$ are the magnitudes of the components of $\delta$ in the normal and tangential direction, respectively.

The correlation between the bonding stress and the opening/sliding displacement under several stress situations is shown in Figure 3 [47]. Figure 3a demonstrates the correlation between the bonding stress in the normal direction and the opening of the adjacent finite elements or the distortion of the four-node joint element in the normal direction, i.e., pure-mode-I fracture. According to Figure 3a, the stress increases with the increase in the opening of the adjacent finite elements $\delta_n$ before it reaches a critical displacement $\delta_{np}$, prescribed by the tensile strength of the material $\sigma_t$. At this stage, no fractures occur. For separation $\delta_n < \delta_{np}$, Equation (3) provides the bonding stress in the normal direction.

$$\sigma_n = \left[ 2 \cdot \frac{\delta_n}{\delta_{np}} - \left( \frac{\delta_n}{\delta_{np}} \right)^2 \right] \cdot \sigma_t \qquad (3)$$

In Equation (3), $\sigma_t$ is the tensile strength of the element; $\sigma_n$ is the bonding stress in the normal direction; $h$ is the size of the particular finite element; $p_0$ is the penalty term at separation $\delta_n = 0$; and $\delta_{np} = 2h\sigma_t/p_0$ is the separation wherein the tensile strength is equivalent to the bonding stress.

Tensile cracks start to appear as soon as the normal separation approaches the critical displacement, i.e., $\delta_n = \delta_{np}$. While the separation of the adjacent element is above $\delta_{np}$, i.e., $\delta_n > \delta_{np}$, the fracture propagates. The two adjacent finite elements entirely break down into two discrete elements as soon as the distance between their surfaces reaches the ultimate opening, i.e., $\delta_{nu}$. In other words, the four-node joint element embodied in the adjacent of the finite elements is removed, and the fracture is completed. Equation (4) is used at this stage to represent the bonding tension for $\delta_n > \delta_{np}$.

$$\sigma_n = f(D) \cdot \sigma_t \tag{4}$$

In Equation (4), $D$ represents the damage index ranging from 0 to 1, while $f(D)$ represents the mechanical damage model's damage function [48].

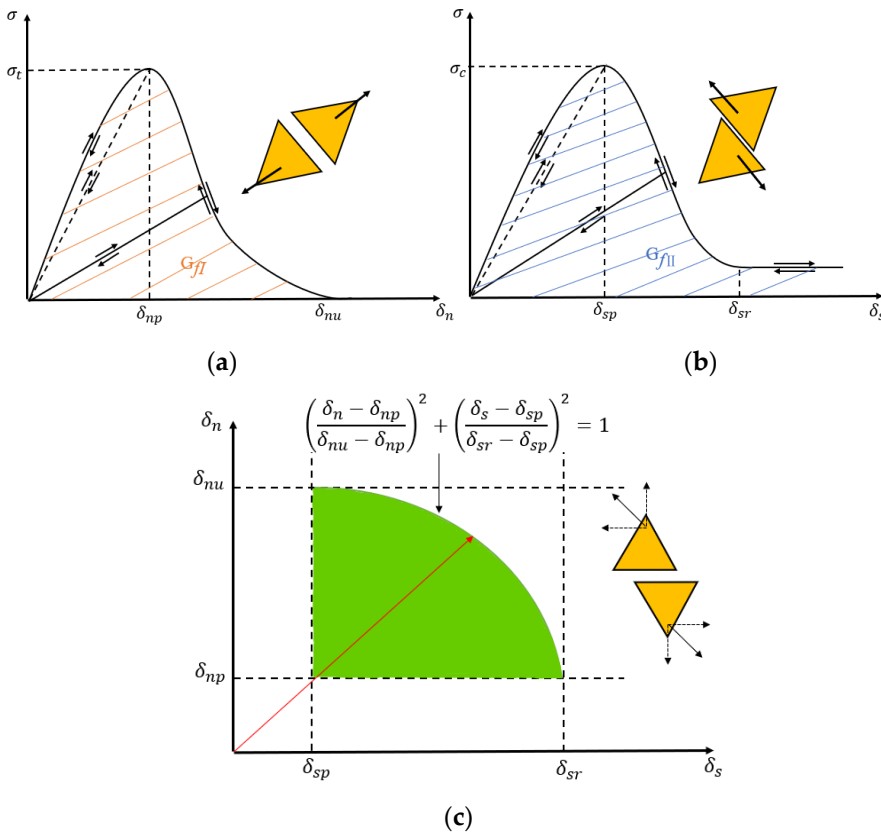

**Figure 3.** Bonding stress and opening/sliding displacement correlation under various stress conditions (after An et al. (2021) [47]): (**a**) under tension conditions; (**b**) under shear conditions; (**c**) under both tension and shear conditions.

The mode-I fracture energy release rate $G_{fI}$, which can be calculated using Equation (5), controls the pure-mode-I fracture process. It is equal to the zone under the curve of the bonding stress and opening displacement, as illustrated in Figure 3a.

$$G_{fI} = \int_{\delta_{np}}^{\delta_{nu}} \sigma_n(\delta_n) d\delta_n \tag{5}$$

In summary, the relationship between the bonding stress in the normal direction and the opening of the adjacent finite elements can be expressed as in Equation (6):

$$\sigma_n = \begin{cases} \left[2\cdot\frac{\delta_n}{\delta_{np}} - \left(\frac{\delta_n}{\delta_{np}}\right)^2\right]\cdot\sigma_t & if \quad 0 \le \delta_n \le \delta_{np} \\ f(D)\cdot\sigma_t & if \quad \delta_{np} \le \delta_n \le \delta_{nu} \\ 0 & if \quad \delta_n \ge \delta_{nu} \end{cases} \tag{6}$$

Figure 3b, which depicts the mode-II fracture model, demonstrates the correlation between the bonding stress and the sliding displacement of two nearby elements or the deformation of the joint element in the tangential direction. The bonding stress in the tangential direction, i.e., the shear stress, can be expressed as in Equation (7) before the sliding displacement $\delta_s$ in the tangential direction reaches a critical slip $\delta_{sp}$, i.e., $\delta_s < \delta_{sp}$.

$$\tau = 2\cdot\frac{\delta_s}{\delta_{sp}}\cdot\sigma_c \tag{7}$$

In this equation, the critical sliding as determined by the shear strength is represented by $\delta_{sp}$, while the shear strength is represented by $\sigma_c$.

Shear failure (pure-mode-II fracture) takes place if the sliding displacement $\delta_s$ in the tangential direction exceeds a critical slip $\delta_{sp}$, i.e., $\delta_s = \delta_{sp}$. When the sliding displacement $\delta_s$ is beyond the critical slip $\delta_{sp}$ described by the tensile strength, i.e., $\delta_s > \delta_{sp}$, the shear failure is assumed to propagate. Equation (8) provides a description of the shear stress at this stage:

$$\tau = g(D)\sigma_c \tag{8}$$

In this equation, $g(D)$ represents the damage functions in the mechanical damage model [48].

Even though the residual opening displacement is greater than the sliding displacement in the tangential direction $\delta_{sr}$, i.e., $\delta_s > \delta_{sr}$, the shear failure is complete. Columb's model defines the bonding stress in the tangential direction as merely frictional resistance. Equation (9) calculates the bonding stress in the tangential direction:

$$\tau = \sigma_n\cdot tan\left(\varnothing_f\right) \tag{9}$$

In this equation, the joint residual friction angle is represented by the symbol $\varnothing_f$.

The zone under the curve of the bonding stress and the sliding displacement, i.e., the energy release rate of the mode-II fracture $G_{fII}$, which is shown in Figure 3b, controls the onset and spread of the fracture during the pure-mode-II fracture process. The $G_{fII}$ is explained by Equation (10):

$$G_{fII} = \int_{\delta_{sp}}^{\delta_{sr}} [\tau(\delta_s) - \tau_r]d\delta_s \tag{10}$$

In summary, the bonding stress in the tangential direction can be calculated according to the sliding of the adjacent finite elements, as described in Equation (11):

$$\tau = \begin{cases} 2\cdot\frac{\delta_s}{\delta_{sp}}\cdot\sigma_c & if \quad 0 \le \delta_s \le \delta_{sp} \\ g(D) & if \quad \delta_{sp} \le \delta_s \le \delta_{sr} \\ \sigma_n\cdot tan\left(\varnothing_f\right) & if \quad \delta_s \ge \delta_{sr} \end{cases} \tag{11}$$

For the mixed-mode-I–II fracture, the correlation between the bonding stress and opening/sliding displacement is shown in Figure 3c. The zone under the curve of the bonding stress and the sliding/opening displacement shown in Figure 3c represents the

fracture energy release rate $G_{fI-II}$ for the mixed-mode-I–II fracture when Equation (12) is satisfied.

$$\left(\frac{\delta_n - \delta_{np}}{\delta_{nu} - \delta_{np}}\right)^2 + \left(\frac{\delta_s - \delta_{sp}}{\delta_{sr} - \delta_{sp}}\right)^2 \geq 1 \tag{12}$$

The dynamic pure-mode-II fracture (shear fracture) is modeled in this study using the Y-HFDEM IDE, which was developed and adapted through the abovementioned methodologies by the authors [2,16,35,36,49]. The authors' earlier improved finite element codes served as the foundation for the implementation of the Y-HFDEM IDE, including TunGeo3D [50], RFPA-RT2D [48], and the open-source combined finite–discrete element libraries Y2D and Y3D, whose original designers were Munjiza (2004) and Xiang et al. (2009), respectively [30,51]. The initial conditions, rock parameters, contact properties, boundary conditions, and fracture criteria could all be set up using the Y-HFDEM IDE to create hybrid FEM/DEM models. The modeled stresses, displacements, velocity, force, damage, fracture, and fragmentation are all displayed in real-time graphs so that faults can be visibly tracked in the proposed method.

### 3. FDEM Modeling of Dynamic Shear Fracture Process

It is worth noting that before carrying on the four-point bending test, the ability and reliability of the FDEM for modeling rock fractures should first be confirmed. Recently, many researchers have validated the proposed method by comparing the modeling results with those obtained from rock mechanics labs and published research. Liu et al. (2016) and An et al. (2020) used the FDEM to model the uniaxial compression strength test and Brazilian tensile strength test, and the acquired results were compared with experiments carried out in a rock mechanics lab and reported in the literature [33,52]. An et al. (2017) modeled a single-borehole blast with a free surface, two-borehole simultaneous blasts, and two-borehole consecutive blasts [34]. They compared the simulation results with those reported in published research, and the reliability of the proposed method was validated [34]. Han et al. (2020) investigated the rock fracture and fragmentation process caused by controlled contour blasting using the FDEM, and the modeling outcomes were consistent with the outcomes of field tests [35]. Their research confirmed that the FDEM has its own merits in handling rock fractures compared with conventional methods. The FDEM can not only model the stress distribution and the initiation and propagation of cracks before rock material fracture, but it can also reflect the interaction between elements during the process of fracture.

In this section, the results of the FDEM-modeled Brazilian tensile strength (BTS) test are compared with results published in the literature. Figure 4 compares the FDEM-modeled results with the experimental results and the typical rock failure process of the BTS test obtained from the literature. The modeled results (Figure 4a) agreed well with the experimental results (Figure 4b), as the main fracture was along the vertical diameter and separated the disc into two halves. Figure 4c shows the typical failure pattern of the BTS test according to the literature [53,54], which included tensile failure along the loading diameter and shear failure at the top and bottom loading areas. Thus, the FDEM can accurately reproduce the rock failure process during the BTS test. The input tensile strength for the FDEM was 10.86 MPa, while the obtained result was 12.87 MPa, which was 1.18 time larger than the input value. Since the loading rate was 0.1 m/s, which is larger than that in the laboratory test, the relatively larger tensile strength could be explained by the effect of the loading rate. Thus, the FDEM can accurately capture the effect of loading on the rock behavior. More information about the calibration of the proposed method can be found in our previously published papers [7,16,31–33,36,47].

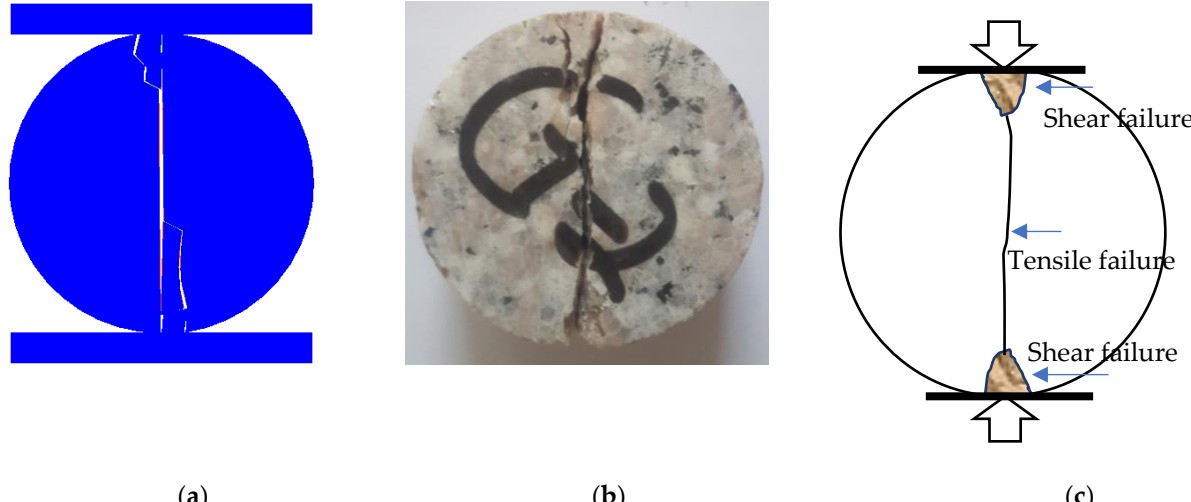

(**a**)             (**b**)             (**c**)

**Figure 4.** Comparison of the FDEM-modeled BTS results with experimental fracture pattern and typical failure pattern results for the BTS test: (**a**) modeled result [33]; (**b**) experimental result; (**c**) typical rock failure pattern of BTS test after Li (2013) and Hobbs (1964) [53,54].

In this section, the FDEM is employed to simulate the rock fracture onset and propagation process during the four-point bending (4 PB) test, which is usually used to demonstrate a pure-mode-II fracture, i.e., a shear fracture. The geometrical model for the 4 PB test is shown in Figure 5a, and the numerical model is shown in Figure 5b. Only the erect sections of the rectangular beam employed for FDEM modeling were taken into consideration, simplifying the problem to plane strain. As shown in Figure 5a, two rigid loading rolls were placed on the top of the specimen, while two rigid rolls were fixed at the bottom of the specimen in both the horizontal and vertical directions. In the four-point bending test, the upper loading rolls move toward the rock specimen in order to provide loading. The toughness of the pure-mode-II fracture can be obtained from the 4 PB test. According to Figure 5a, Equation (13) can be used to evaluate the toughness of a mode-II fracture (Rao, 1999) [55]:

$$K_{IIC} = \frac{P_{Max}}{B\sqrt{D}} \left[ \frac{L_2 - L_1}{L_2 + L_1} \right] \left[ 1.44 - 5.08 \left( \frac{a}{D} - 0.507 \right)^2 \right] \sec \left[ \frac{\pi a}{2D} \right] \sqrt{sin \left[ \frac{\pi a}{2D} \right]} \quad (13)$$

In Equation (3), $L_1$ is the distance between the prefabricated notch and the supporting point $A$; $L_2$ is the distance between point $B$ and $C$; $P_{Max}$ is the peak load; $a$ is the length of the prefabricated notch; $D$ is the width of the rectangular beam; $B$ is the thickness of the rectangular beam; and $K_{IIC}$ is the mode-II fracture toughness.

Figure 5b depicts the numerical models for the 4 PB test. It can be seen that triangle elements were applied to discretize the models. The rock parameters can be found in Table 1. During this test, four dynamic loading rates, i.e., 1 m/s, 5 m/s, 10 m/s, and 50 m/s, were applied on the top two rigid rolls to study the crack onset and spread of mode-II (shearing) fracturing under various loading rates.

**Table 1.** Rock parameter for the FDEM.

| Symbol | Parameter | Value | Unit |
|:---:|:---:|:---:|:---:|
| $E$ | Young's modulus | 60 | GPa |
| $\nu$ | Poisson's ratio | 0.26 | N/A |
| $\rho$ | Density | 2600 | Kgm$^{-3}$ |
| $\sigma_t$ | Tensile strength | 20 | MPa |

**Table 1.** *Cont.*

| Symbol | Parameter | Value | Unit |
|--------|-----------|-------|------|
| $\sigma_c$ | Compressive strength | 200 | MPa |
| $\varnothing$ | Internal friction angle | 30 | ° (Degree) |
| $u$ | Surface friction coefficient | 0.1 | − |
| $G_{fI}$ | Mode-I fracture energy release | 50 | $Nm^{-1}$ |
| $G_{fI}$ | Mode-II fracture energy release | 250 | $Nm^{-1}$ |

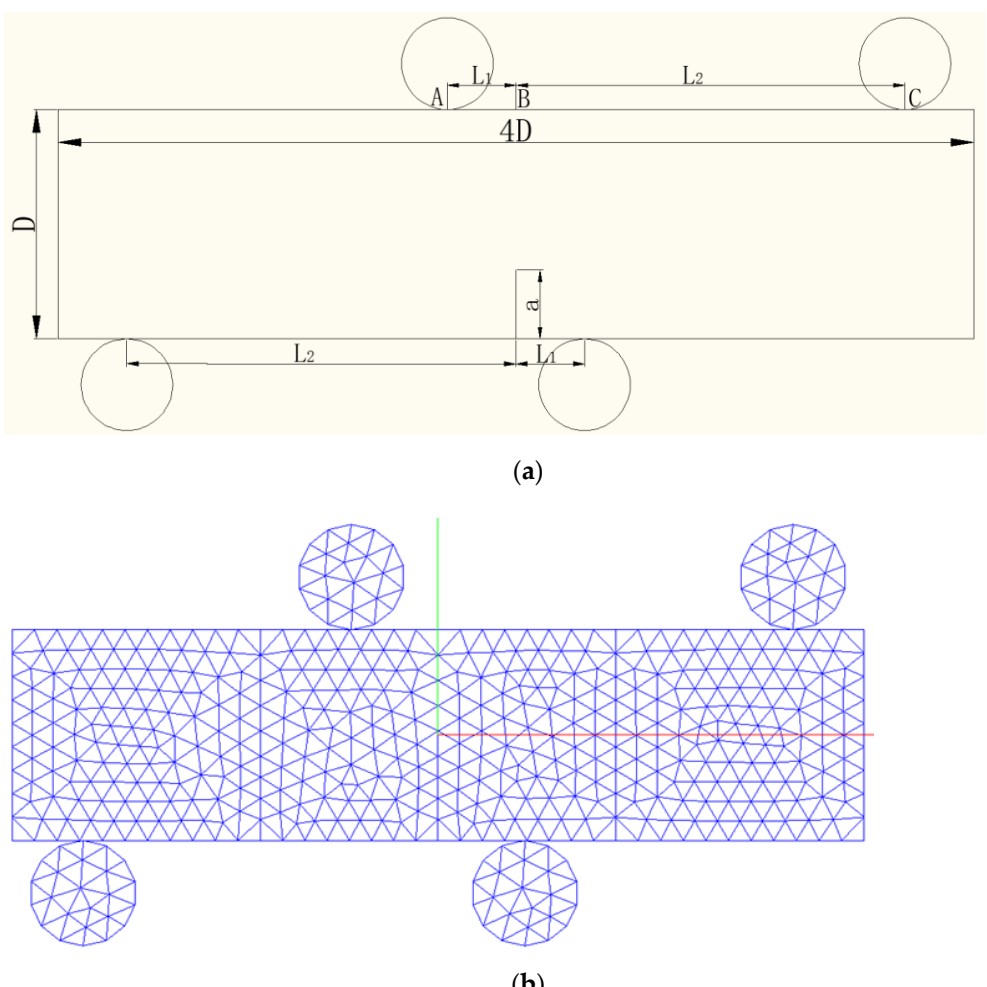

**(a)**

**(b)**

**Figure 5.** Geometrical and numerical models for four-point bending test. (**a**) Geometrical model; (**b**) numerical model.

### 3.1. Under the Loading Rate of 1 m/s

Figure 6 shows how the stress spread during the 4 PB test, and Figure 7 shows how the cracks began and progressed. On top of the beam, a continuous displacement increase of 1 m/s for each rigid roll was applied during the 4 PB test process. The force-loading displacement curve is depicted in Figure 8a, while the relationships between the force-loading and the crack opening/sliding displacements (CMOD/CMSD) are illustrated in Figure 8b,c, respectively. The letters in Figure 8 correspond to those in Figure 6.

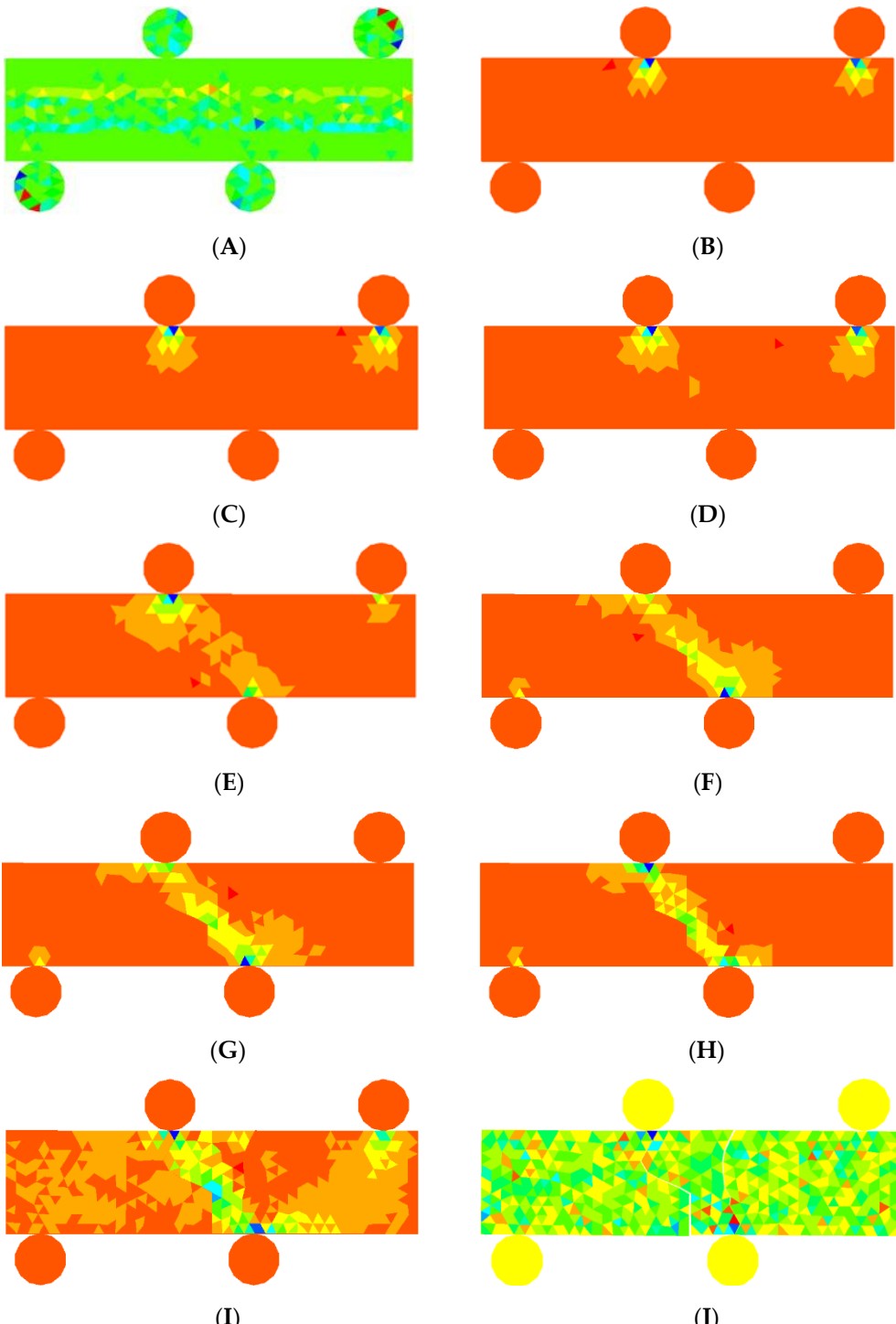

**Figure 6.** Stress spread during the 4 PB test under quasi-static loading modeling by FDEM. (**A**) 0 mm (0 ms), (**B**) 0.0025 mm (0.0025 ms), (**C**) 0.0075 mm (0.0075 ms), (**D**) 0.015 mm (0.015I), (**E**) 0.05 mm (0.05 ms), (**F**) 0.075 mm (0.075 ms), (**G**) 0.08 mm (0.08 ms), (**H**) 0.1 mm (0.1 ms), (**I**) 0.14 mm (0.14 ms), (**J**) 0.38 mm (0.38 ms).

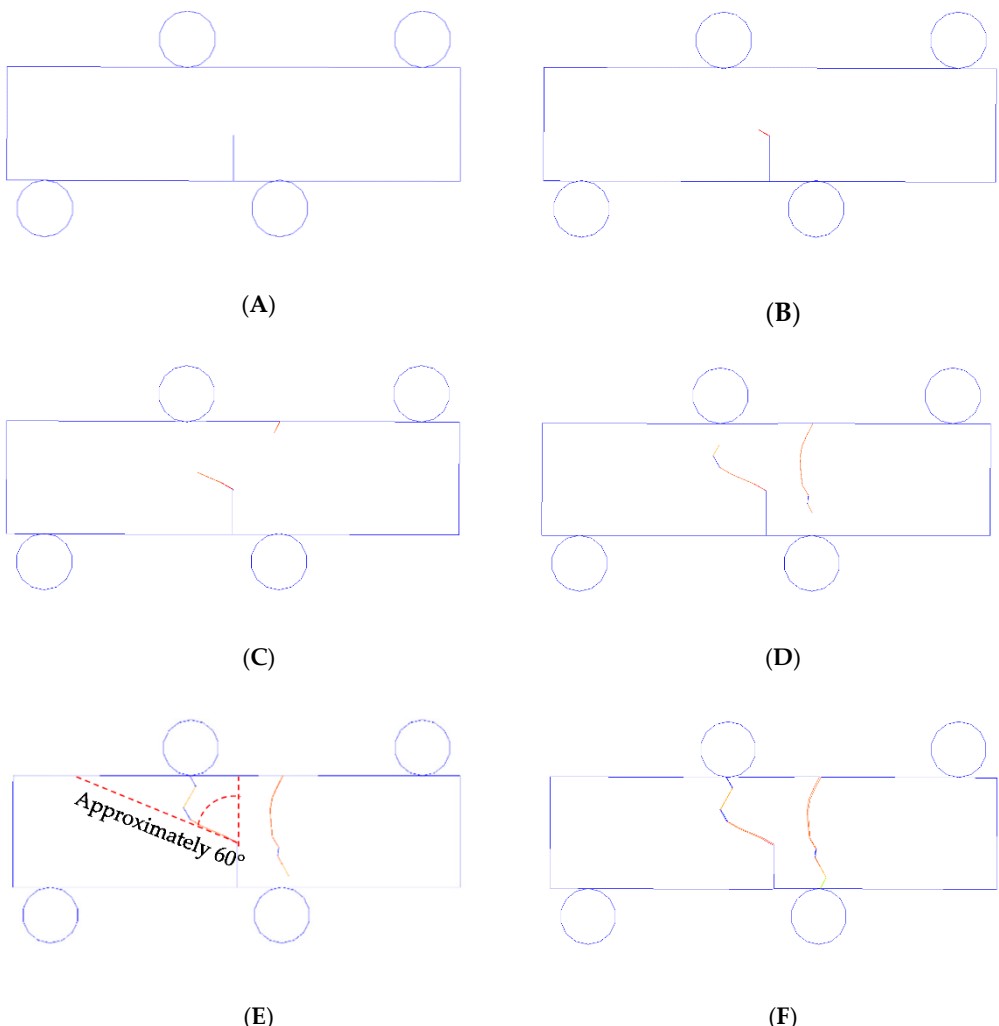

**Figure 7.** Crack initiation and progression during the 4 PB test under a constant 1 m/s loading rate. (**A**) 0 mm (0 ms), (**B**) 0.0075 mm (0.0075 ms), (**C**) 0.08 mm (0.08 ms), (**D**) 0.11 mm (0.11 ms), (**E**) 0.14 mm (0.14 ms), (**F**) 0.38 mm (0.38 ms).

First, the midpoint of the bottom beam was marked with a prefabricated notch (Figure 5a). As the rigid two rolls on the top of the beam moved downwards and touched the beam, stresses were immediately produced in the vicinity of the points of contact between the specimen and the two loading rolls (Figure 6B). The strain was mostly concentrated in the loading vicinity while the two loading rolls kept moving (Figure 6C); subsequently, the stress concentration at the tip of the prefabricated notch could be observed (Figure 6D). Next, the stress was mainly distributed along the line connecting the top and bottom rolls near the middle beam. During this period, even though fluctuation occurred, the loading force rose quickly (Figure 8a(A,B)). Almost no sliding took place (Figure 8a(A,B)), while the prefabricated notch expanded gently (Figure 8a(A,B)). Subsequently, a crack appeared, starting at the prefabricated notch's tip, and spread to the left loading point (Figure 7B). Because the crack started from the prefabricated notch, the points of contact between the beam and the rigid rolls were not very tight. Thus, the force suddenly dropped. However, the beam did not completely lose its ability to carry loads. As the rigid rolls continued to move, the force continually increased. A new crack was produced at the top of the beam between the two loading areas as the existing crack continued to spread (Figure 7C). The force rose quickly until it reached its maximum (Figure 8a(D)) when the two cracks spread (Figure 7D). The CMOD and CMSD also increased in the meantime (Figure 8b(B–D) and Figure 8c(B–D)). The force-loading displacement curve then started to rapidly decline

(Figure 8a(D,E)), since the crack generated from the prefabricated notch' tip approached the left loading point (Figure 7E). Eventually, the beam completely lost the ability to carry loads (Figure 8a(E,F)), while the crack mouth's maximum opening/sliding gaps were achieved.

The force-loading displacement curve had two summits, as shown in Figure 8a. The first summit was caused by cracks that started from the tip of the prefabricated notch, while the second peak was due to the crack initiated from the top of the beam. Because the cracks that started from the tip of the prefabricated notch caused the first peak of the force-loading displacement curve, the first peak force (1.21 MN) was used to calculate the pure-mode-II fracture toughness according to Equation (14):

$$K_{IIC} = \frac{P_{Max}}{B\sqrt{D}} \left[ \frac{L_2 - L_1}{L_2 + L_1} \right] \left[ 1.44 - 5.08 \left( \frac{a}{D} - 0.507 \right)^2 \right] \text{Sec} \left[ \frac{\pi a}{2D} \right] \sqrt{\sin \left[ \frac{\pi a}{2D} \right]} = 4.979 \text{Mpa} \sqrt{m} \quad (14)$$

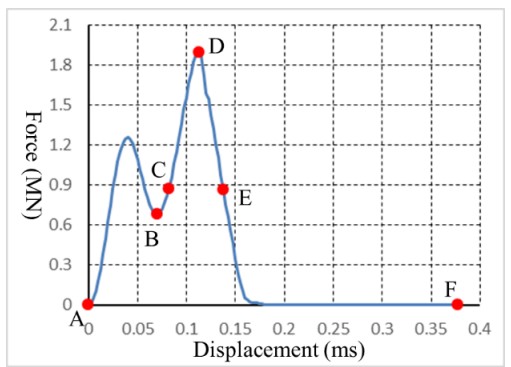

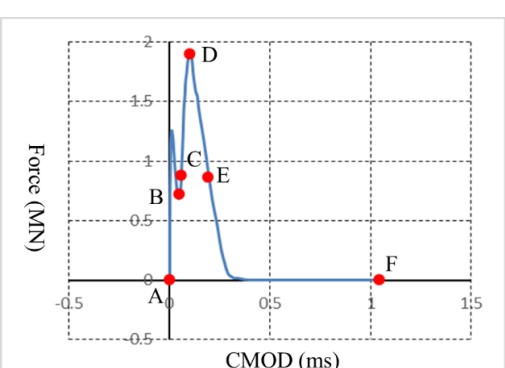

(a)

(b)

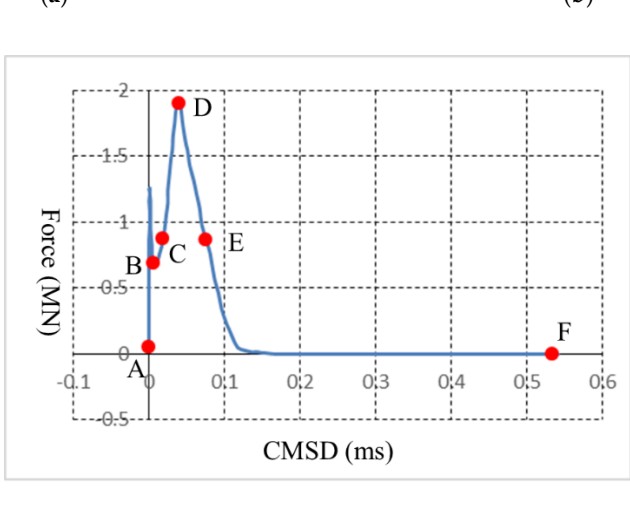

(c)

**Figure 8.** Force−loading−related curves for the 4 PB test under a loading rate of 1 m/s. (**a**) Force−loading displacement curve; (**b**) force−loading CMOD curve; (**c**) force−loading CMSD curve. (**A**−**D**) loading force rose; (**D**) loading force reached the maximum; (**D**−**E**) loading force declined; (**F**) the beam lost its load-carrying ability.

### 3.2. Under the Loading Rate of 5 m/s

Since the stress propagation processes for the 4 PB tests under the loading rate of 5/s, 10 m/s, and 50 m/s were quite similar, the stress propagation under these three loading rates is not examined in the following sections.

Figure 9 visually illustrates the crack initiation and spread in the 4 PB test under a loading rate of 5 m/s, while Figure 9 shows the corresponding force-loading displacement curve and force-loading CMOD and CMSD curves. The letters in Figure 10 correspond

to those in Figure 9. As the two upper loading rolls made contact with the beam, compressive stress from each of the loading rolls initiated immediately and increased sharply (Figure 10a(A,B)). Because of the concentration of compressive stress in the loading vicinity of the loading rolls and the beam, shear cracks occurred as the concentration of high compressive stress reached the compressive strength (Figure 9B). With the continuous descent of the loading rolls, cracks from the two loading contact points continued to propagate (Figure 9C). Meanwhile, a crack initiated from the tip of the prefabricated notch and spread to the top-left loading area (Figure 9C). When the crack initiating from the prefabricated notch reached the top-left loading area, the cracks initiating from the top-right loading point reached the bottom of the beam (Figure 9D). As the loading rolls kept moving, a crack initiated from the top-left loading point and propagated to the left-bottom fixing roll (Figure 9E). The final fracture pattern included two cracks initiating from the two top loading points, one crack initiating from the prefabricated notch, and several fractures at the two top loading points (Figure 9F).

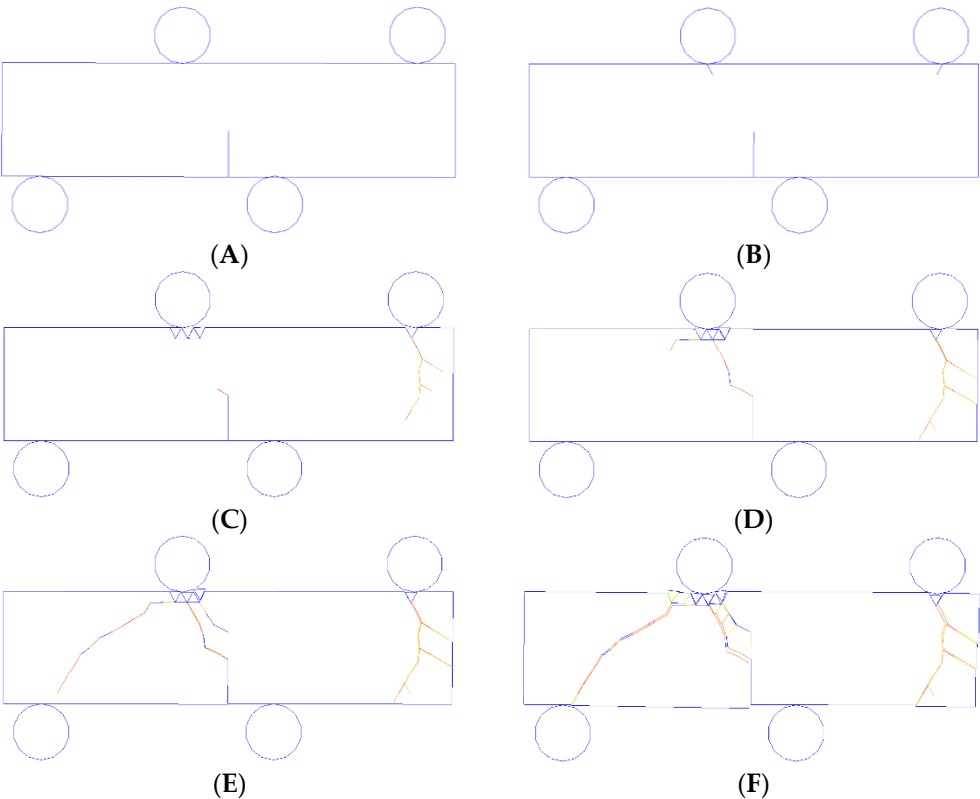

**Figure 9.** Crack beginning and progression during the 4 PB test under a constant 5 m/s loading rate. (**A**) 0 mm (0 ms), (**B**) 0.075 mm (0.0015 ms), (**C**) 0.125 mm (0.025 ms), (**D**) 0.1875 mm (0.0375 ms), (**E**) 0.325 mm (0.065 ms), (**F**) 0. 9375 mm (0.1875 ms).

Figure 10a indicates that a brittle material failure process was captured by the hybrid finite–discrete element method. The sharp stress increase was related to the impact of the loading rolls on the beam and indicates the stress concentrations at the two loading points (Figure 10a(A,B)). The sudden drop in stress (Figure 10a(B,C)) was due to the fractures that occurred at the loading points, which made the beam load bearing decrease (Figure 10a(B,C)). However, the loading capability was not completely lost (Figure 10a(C)). As the cracks that initiated from one side of the beam reached the other side of the beam and the beam was divided into several pieces by the long cracks (Figure 9E,F), the beam's load-carrying ability was completely lost (Figure 10a(F)).

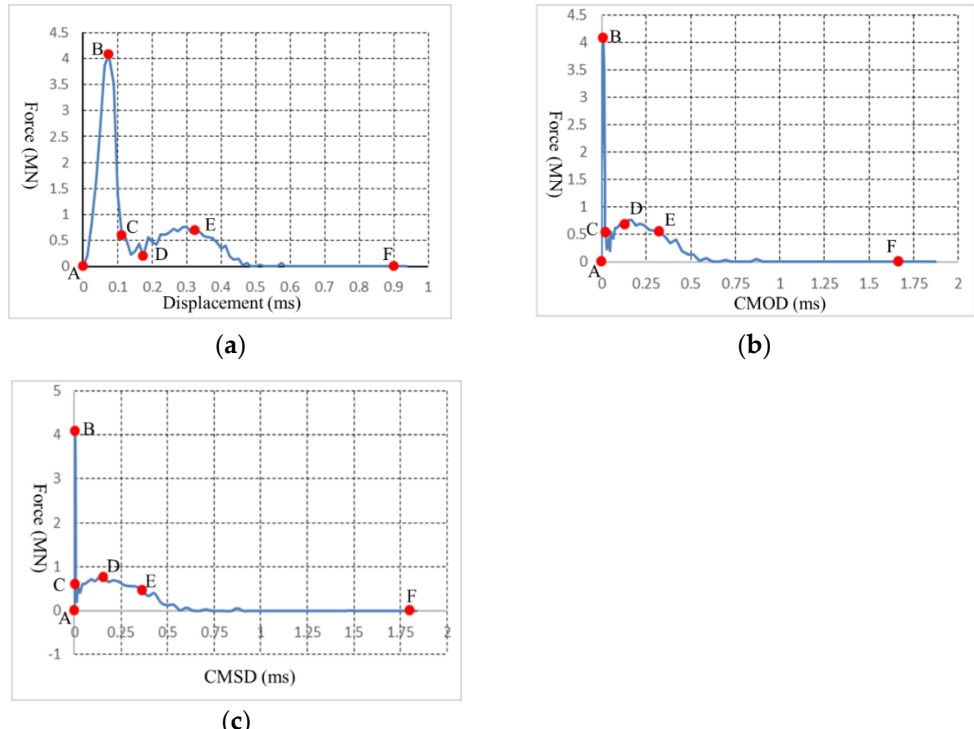

**Figure 10.** Force−loading−related curves for the 4 PB test under a loading rates of 5 m/s. (**a**) Force−loading displacement; (**b**) force−loading CMOD curve; (**c**) force−loading CMSD curve. (**A**−**B**) loading force increased; (**B**) loading force reached maximum; (**B**−**E**) loading force declined; (**F**) the beam lost its load−carrying ability.

Figure 10b,c illustrate the relationship between the force loading and the CMOD and CMSD. The CMOD and CMSD were mainly associated with the crack propagation. It can be seen that although the force increased and dropped sharply, the CMOD and CMSD demonstrated almost no increment (Figure 10b(A–C) and Figure 10c(A–C)), as no cracks occurred at the very beginning (Figure 9AB). As the crack initiated from the prefabricated notch's tip (Figure 9C), the CMOD and CMSD began to increase rapidly, while a relatively low force fluctuation occurred (Figure 10b(C–E) and Figure 10c(C–E)) as the beam lost its ability to carry loads. Finally, the CMOD and CMSD continued to increase without the application of force on the beam as the beam completely lost its bearing capacity (Figure 10b(E,F) and Figure 10c(E,F)).

The peak force (4 MN) was used to calculate the pure-mode-II fracture toughness according to Equation (15):

$$K_{IIC} = \frac{P_{Max}}{B\sqrt{D}}\left[\frac{L_2 - L_1}{L_2 + L_1}\right]\left[1.44 - 5.08\left(\frac{a}{D} - 0.507\right)^2\right]\sec\left[\frac{\pi a}{2D}\right]\sqrt{\sin\left[\frac{\pi a}{2D}\right]} = 16.459\text{Mpa}\sqrt{m} \quad (15)$$

### 3.3. Under the Loading Rate of 10 m/s and 50 m/s

The rock fracture and fragmentation process in the 4 PB test at a loading rate of 10 m/s is depicted in Figure 11. Due to the large loading rate (10 m/s) of the top rolls impacting on the beam, cracks first appeared at the two upper loading areas (Figure 11B). The loading points were smashed into fragments due to the high stress concentration (Figure 11C). Then, a crack initiated from the tip of the prefabricated notch and spread to the top-left loading point, while two long cracks began from the loading points and propagated approximately to the bottom-right fixing point (Figure 11D). Finally, the crack from the top-left loading point met the crack from the prefabricated notch, while the cracks from the top-right loading point reached the bottom-right fixing point (Figure 11E,F).

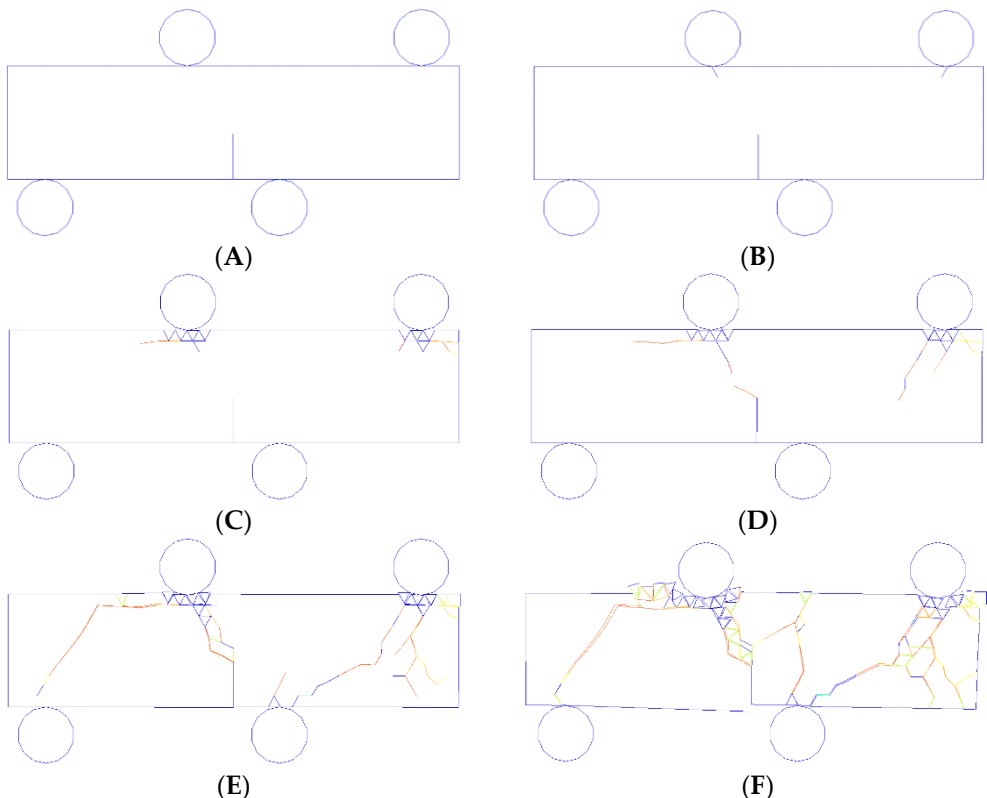

**Figure 11.** Crack initiation and progression during the 4 PB test under a constant 10 m/s loading rate. (**A**) 0 mm (0 ms), (**B**) 0.075 mm (0.0075 ms), (**C**) 0.15 mm (0.015 ms), (**D**) 0.2 mm (0.02 ms), (**E**) 0.5 mm (0.05 ms), (**F**) 2.4 mm (0.24 ms).

Figure 12 illustrates the fracture and fragmentation process for the 4 PB test under a loading rate of 50 m/s. Due to the large impact from the loading rolls, fractures began at the loading points and propagated to the bottom of the beam. As the beam was smashed into fragments, the stress distribution on the tips of the prefabricated notch did not meet the rock strength. Thus, no cracks initiated from the tip of the prefabricated notch. The fracture was dominated by shear failure and was mainly distributed at the two top loading points (Figure 12E,F).

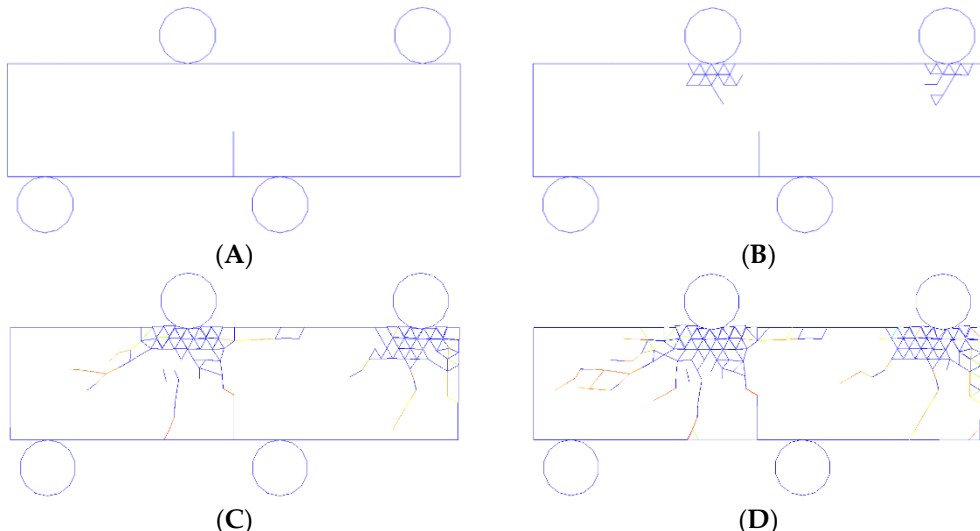

**Figure 12.** *Cont.*

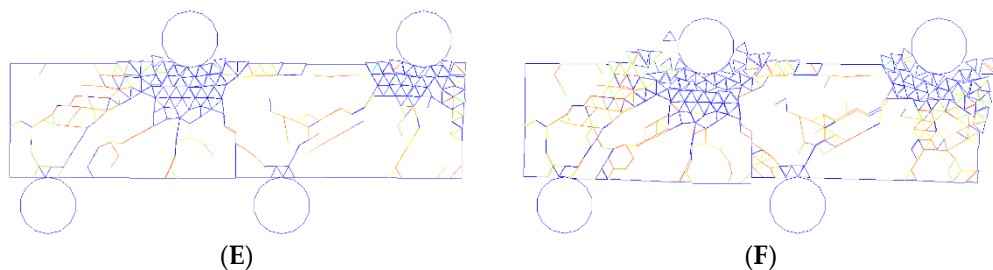

(**E**)          (**F**)

**Figure 12.** Crack initiation and progression during the 4 PB test under a constant 50 m/s loading rate. (**A**) 0 mm (0 ms), (**B**) 0.25 mm, (0.005 ms), (**C**) 0.75 mm (0.015 mI (**D**) 1 mm (0.02 ms), (**E**) 2 mm (0.04 ms), (**F**) 5 mm (0.10 ms).

## 4. Discussion

### 4.1. Effect of Loading Rate on the Rock Behaviors

For rock-like materials, the loading rate significantly influences the material behaviors, e.g., stiffness, fracture pattern, and fracture path [56–58]. Therefore, the hybrid finite–discrete element method takes the effect of the loading rate into account to naturally simulate rock fracture behavior under various loading conditions. The correlation between the uniaxial compressive strength and the loading rate (Equation (16)) was implanted into the hybrid finite–discrete element method in order to take the impact of the loading rate on the rock behavior into consideration. The correlation between the tensile strength and the loading rate can be determined using Equation (16), which was based on various dynamic uniaxial and triaxial compression, uniaxial tension, and unconfined shear experiments on Bukit Timah granite [56].

$$\sigma_{cd} = A \cdot log\left(\frac{\dot{\sigma}_{cd}}{\dot{\sigma}_c}\right) + \sigma_c \tag{16}$$

In Equation (16), $A$ is the material parameter, which was determined to be 11.9 from the Bukit Timah granite experiments [56]; $\sigma_c$ is the uniaxial compressive strength under the quasi-static loading rate (MPa); $\dot{\sigma}_c$ is the quasi-static loading rate (roughly $5 \times 10^{-2}$ MPa/s); $\sigma_{cd}$ is the dynamic uniaxial compressive strength (MPa); and $\dot{\sigma}_{cd}$ is the dynamic loading rate (MPa/s).

The fracture process is controlled by the fracture energy release rate rather than the fracture toughness in the hybrid finite–discrete element method. This study supports the hypothesis that the fracture energy release rate rises according to Equation (16).

The FDEM obtained pure-mode-II fracture toughness values of 4.979 Mpa$\sqrt{m}$ and 16.459 Mpa$\sqrt{m}$ under the loading rates of 1 m/s and 5 m/s, respectively. Obviously, the fracture toughness for the loading rate of 5 m/s was much higher than that under the loading rate of 1 m/s, which indicates that the loading rate significantly influences rock behaviors.

The Brazilian tensile strength (BTS) tests were simulated with loading rates of 0.05 m/s, 0.1 m/s, 0.25 m/s, 0.5 m/s, 0.75 m/s, 1 m/s, 2.5 m/s, 5 m/s, and 10 m/s in order to show the FDEM's capacity to capture the effect of the loading rate on rock strength. Several representative examples of the simulation results are provided in Figure 13.

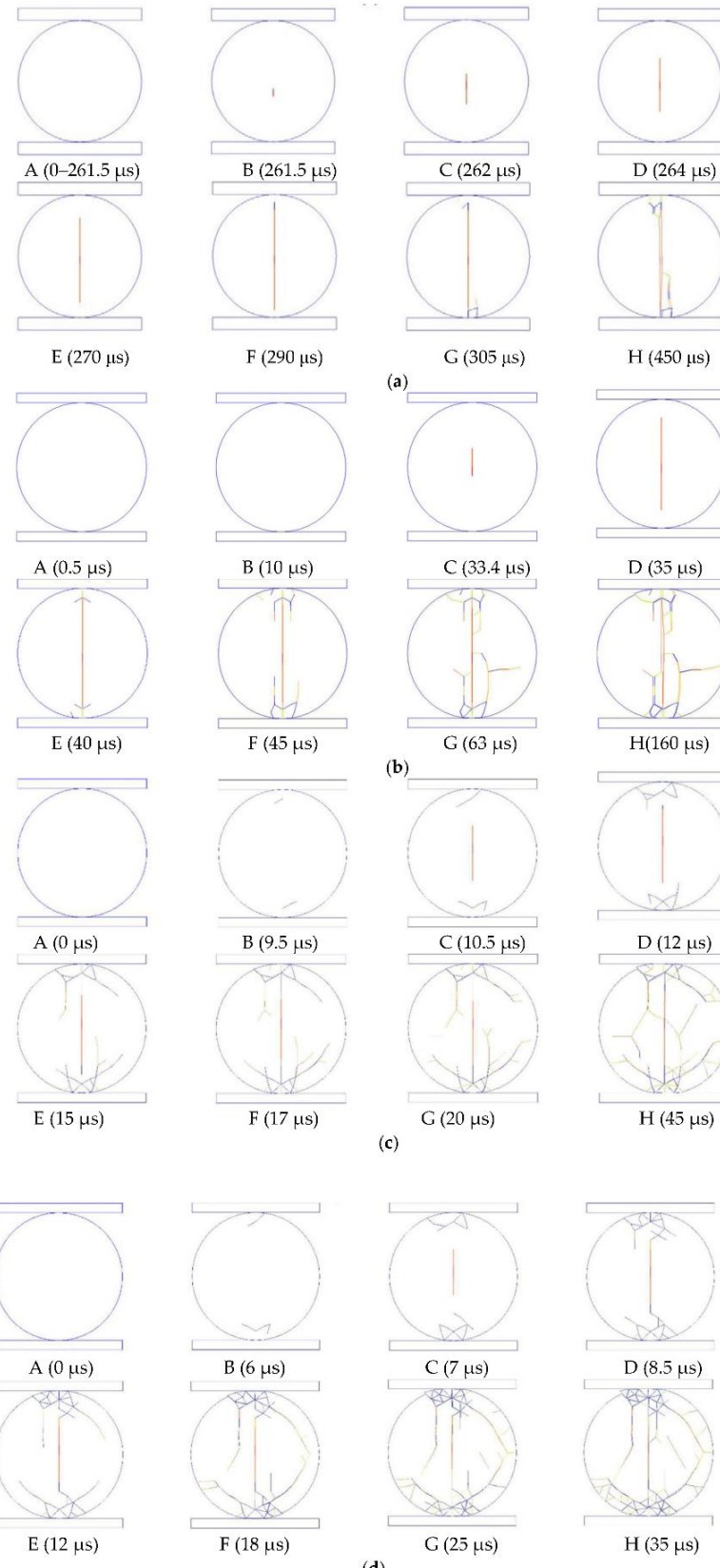

**Figure 13.** Crack initiation and progression during the BTS test under different loading rates. (**a**) 0.1 m/s; (**b**) 1 m/s; (**c**) 5 m/s; (**d**) 10 m/s.

In order to obtain accurate results for the effect of the loading rate on rock behavior, nine groups of BTS tests were modeled under various loading rates in total. It would be cumbersome to show all the simulation results here, so only four groups of representative results (under loading rates of 0.1 m/s, 1 m/s, 5 m/s, and 10 m/s) are illustrated in Figure 13. It can be seen that the initiation and progression of the cracks as well as the rock fragmentation varied according to the loading rate. To more effectively compare the simulation results with those reported in the literature, the acquired tensile strength was used to calculate the dynamic strength increasing factor (DIF), which is the ratio between the dynamic and static strength. In Figure 14, the DIF is displayed on the vertical axis, the strain rate's logarithm is displayed on the horizontal axis, and the experimental results and FDEM-modeled results are both shown. As indicated in Figure 14, the FDEM-modeled results demonstrated the same trend as those obtained from the literature for an increasing loading rate. In general, the results modeled by the proposed method agreed well with the experimental findings in the literatures [59,60], which indicated that the influence of the loading rate was accurately reflected by the FDEM through the implementation of the correlation between the rock strength and loading rates.

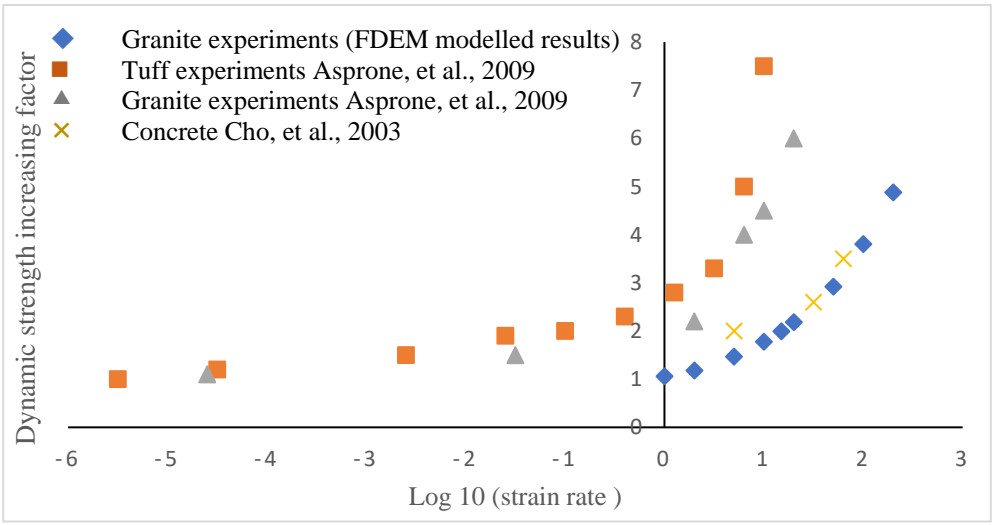

**Figure 14.** Comparison of the FDEM−modeled DIF with the results from the literatures [59,60].

### 4.2. Comparison of the FDEM-Modeled Fractures with Those Reported in the Literature

Liu, Kou, et al. (2007) modeled a four-point bending test using a numerical tool, R-T2D [6]. The geometrical model used in their study is the same as that used in the present study. The obtained results are illustrated in Figure 14. As this test was conducted under static loading conditions, the results obtained using R-T2D are only compared with the FDEM-modeled results under quasi-static loading, i.e., under the loading rate of 1 m/s.

The modeling results obtained by the FDEM in the present study (Figure 7) agreed well with those obtained by R-T2D (Figure 15). According to the abovementioned study, the crack initiated from the tip of the prefabricated notch and then propagated to the left loading point on the top surface of the rock specimen, as illustrated in Figure 15. Rao (1999) studied mode-II fracture toughness using a four-point bending test through both experimental and numerical approaches [55]. The experimental results observed by Rao (1999) pointed towards the same conclusion, that the crack initiated from the tip of the prefabricated notch and nearly reached the left loading point on the top surface of the rock specimen. Additionally, Rao (1999) concluded that the fracture angle was around 60°. In our research, the fracture angle was also approximately 60°, as indicated in Figure 7E. Thus, the FDEM-modeled fracture pattern was similar to that observed by Rao (1999) [55] and Liu, Kou, et al. (2007) [6].

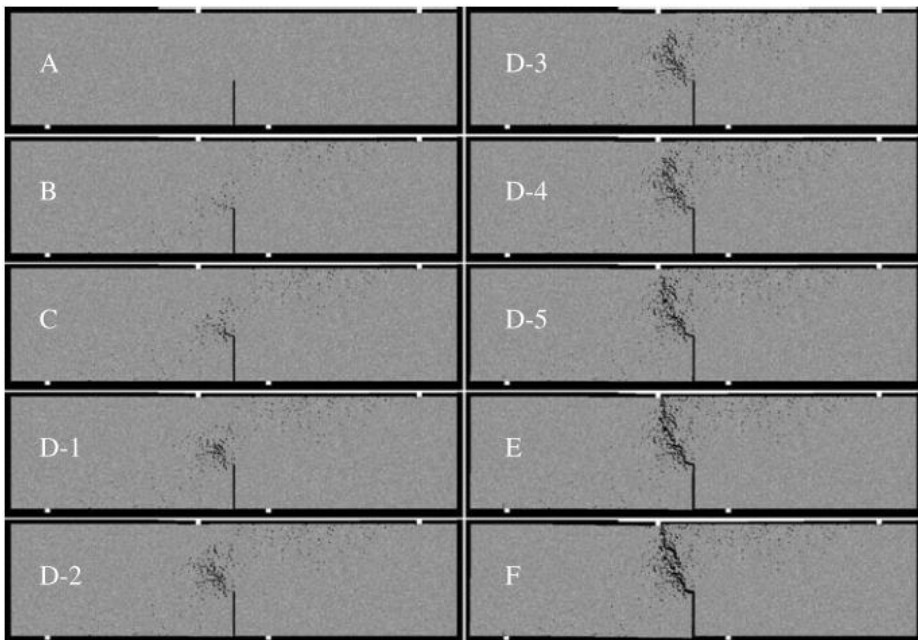

**Figure 15.** Fracture progression process in the four-point bending test modeled using R-T$^{2D}$ [6] (**A**) no crack; (**B**) crack initiated in front of the notch tip; (**C**) crack initiation angle is about 60°; (**D-1**–**D-5**) fracture propagated; (**E**) crack reached the edge of the specimen; (**F**) specimen was split into two halves.

## 5. Conclusions

In this study, a hybrid finite–discrete element method was proposed to model the pure-mode-II fracture process under dynamic loading conditions. The hybrid finite–discrete element method combines the benefits of the discrete element method for simulating interactions and solid fracturing with those of the finite element approach for describing elastic deformations. To model the transition from continuum to discontinuum, three fracture models were proposed to simulate the rock fracture process. The proposed method outperformed the conventional continuum-based finite element method and discontinuum-based discrete element method due to the successful modeling of the continuum to discontinuum transition through fracture and fragmentation. Then, the hybrid finite–discrete element method was implemented to model the pure-mode-II fracture process using a four-point bending test under dynamic loading conditions. During the four-point bending tests, the hybrid finite–discrete element method accurately modeled the stress propagation as well as the initiation and spread of cracks. Since it is generally agreed that rock behavior is significantly impacted by the loading rate, the hybrid finite–discrete element method incorporated an empirical correlation between the static and dynamic strength acquired from dynamic rock fracture tests, so that the impact of the loading rate could be considered. Subsequently, the influence of the loading rate on the rock fracture toughness and rock strength were discussed. The significance of this study can be summarized as follows:

The hybrid finite–discrete element method combines the advantages of both continuum and discontinuum methods so that it can accurately model the pure-mode-II fracture process during a four-point bending test. During the 4 PB test, while under a lower loading rate (e.g., 1 m/s and 5 m/s), a fracture first initiated from the tip of the prefabricated notch and propagated to the loading point at the top surface of the beam. However, for relatively higher loading rates (e.g., 10 m/s and 50 m/s), the fracture began to appear at the bottom middle of the beam.

The force-loading displacement curves from the 4 PB test simulation depicted a typical process of rock failure and illustrated that the hybrid finite–discrete element is capable of accurately capturing the dynamic fracture behavior under different dynamic loads through

applying an empirical correlation between the static and dynamic strength acquired from dynamic rock fracture tests.

**Author Contributions:** Conceptualization, H.L., H.A. and Y.S.; investigation, H.L., Y.S. and H.A.; writing—original draft preparation, H.A., Y.S. and S.W.; funding acquisition, Y.F. and H.A. All authors have read and agreed to the published version of the manuscript.

**Funding:** This work was partly supported by funding from the Research Center for Analysis and Measurement KUST (Analytic and Testing Research Center of Yunnan, grant number 2020T20180040 and 2021M20202139010); the Guizhou Provincial Department of Education Natural Science Research Top-notch Talents Project (Y[2020]041); and Guizhou High-level innovative talents training project (2016-21), for which we are greatly appreciative.

**Institutional Review Board Statement:** Not applicable.

**Informed Consent Statement:** Not applicable.

**Data Availability Statement:** The data used to support the findings of this study are included in the article.

**Conflicts of Interest:** The authors declare that they have no conflict of interest regarding the publication of this paper.

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
