# Peer review of "Investigation of the Dynamic Pure-Mode-II Fracture Initiation and Propagation of Rock during Four-Point Bending Test Using Hybrid Finite–Discrete Element Method"

_sustainability, doi:10.3390/su141610200_

Round 1

Reviewer 1 Report

The authors studied the shear fracture of rocks at different loading rates from 1-50 m/s using the hybrid finite-discrete element method. This paper seems like a technical report instead of a scientific paper. My comments on this manuscript are listed as follows:

1. What’s the novelty of this manuscript? Did the authors propose new numerical methods? If yes, what’s the advantage of the new numerical method? Did the authors find something new results using this method? What’s the comparison of the new method to the traditional numerical method?

2. In addition to the numerical modeling, what’s the experimental evidence of the crack initiation and progression at a high constant loading rate?

3. Page 7, Table 1, where do the tensile strength 20 MPa and compressive strength 200 MPa come from? So, the strength values are independent of the loading rate in the modeling part? In addition, how do the authors know the surface friction coefficient of 0.1?

4. Section 4, what kind of simulations did the authors use to study the effect of loading rate on the rock behaviors? Compression, tension, or bending? Pls give an image of the model.

5. Page 15, Line 472, The FDEM obtained pure mode-II fracture toughness under the loading rate of 1m/s 472 and 5m/s are 4.979Mpa? and 16.459Mpa? , respectively. What are the fracture toughness values at 10 m/s and 50 m/s?

6. Page 15, equation 16, the compressive strength is used. How do the authors know that tensile strength and shear strength follow the same equation? By the way, how to evaluate the shear strength of rock? How does the shear strength affect the current numerical simulation of dynamic shear fracture of rocks?

In general, this manuscript doesn’t show a rigorous characteristic. This manuscript is not suitable for publication unless the above comments are addressed.

Author Response

Dear Reviewer

Thanks very much for reviewing our manuscript and giving us constructive suggestion and comments. We have revised our manuscript according to your review and comments.

The following is a summary of changes against each point which is being raised by you. In the following, the paper submitted for review in last time is referred here as the original manuscript and this time is referred as revised manuscript. Two copies of the revised manuscripts are to be submitted in this time in one PDF file, the first part of the PDF file is the marked copy, where the changes are marked using the function of MS Word`s Track Changes, while the second part is the normal copy of the main manuscript. It is worth noting that the number of lines mentioned below is based on the marked copy.

In the following, we firstly list the review comments/suggestions which are raised by your then present our responses. When presenting our responses, we firstly indicate whether we agree/disagree. We then explain how the changes are made to incorporate your review comments and where are the changes in the marked copy of the revised manuscript if we agree your comments. Otherwise, the reason is to be explained.

Comments made by Reviewer

The authors studied the shear fracture of rocks at different loading rates from 1-50 m/s using the hybrid finite-discrete element method. This paper seems like a technical report instead of a scientific paper. My comments on this manuscript are listed as follows:

Authors response: Thank you very much for the comments.

It might seem like a technical report. However, this is our research article, as we developed a numerical tool and use it to investigate the rock fracture under impact loadings.

Specific comments:

  1. Whats the novelty of this manuscript? Did the authors propose new numerical methods? If yes, whats the advantage of the new numerical method? Did the authors find something new results using this method? Whats the comparison of the new method to the traditional numerical method?

Authors’ response: Thanks for the comments and constructive suggestions.

The FDEM has been proposed by the authors to model the conventional rock mechanical tests, e.g. Brazilian dis test[1], Uniaxial compressive test[2], and implemented in many engineering problems, e.g. tunnel excavation[3], slope failure analysis[4]. The previous studies focus on the macro phenome of the rock mechanism, while this research try to illustrate the micro phenomenon of the rock fracture process. This paper aims at demonstrating the capabilities of the FDEM in modelling the dynamic behavior throughout the course of pure mode-II fracture and the combined method's capacity to accurately catch how loading rate affects dynamic fracture toughness. To successfully model the pure mode-II fracture process, the transition from continuum to discontinuum, the three fracture modes and the effect of the loading rate are considered.

The novelties of the FDEM are described in detail as follows. 

Firstly, a hybrid finite-discrete element method is proposed and developed on the basis of the previous achievement in literature to overcome the limitations of the continuum method (e.g. finite element method) or discontinuum method (e.g. discrete element method) in modelling the complete rock fracture and fragmentation process. The hybrid finite-discrete element method takes the advantages of the both the continuum methods and discontinuum methods and can model the transition from continuum to discontinuum. Then, an integrated development environment is development for the hybrid finite-discrete element method, i.e. Y-HFDEM. The Y-HFDEM is development using C++ and OpenGL on the basis of the author`s 2D and 3D on the basis of their previous enriched finite element codes RFPA-RT2D [5] and TunGeo3D [6], and the open-source combined finite-discrete element libraries Y2D and Y3D originally developed by Munjiza (2004)[7] and Xiang et al. (2009) [8], respectively.

Secondly, the authors proposed three fracture models for model the Mode-I (tensile failure), Mode-II (shear failure) and mixed mode I-II fractures. The combined single and smeared crack model was developed by Munjiza et al. (1999) for modelling tensile failure (Mode-I) of concrete only in the Y2D/ 3D open-source. The Y-HFDEM extends the combined single and smeared crack model for modeling various fracture models by proposing three fracture models. Thus, the hybrid finite-discrete element method not can only model tensile failures but also can model shear failures and mixed fractures.

Thirdly, the Y-HFDEM takes the effect of loading rate into account by implementing an empirical relation between the static strengths and the dynamic strengths derived from the dynamic rock fracture experiments (From Line 127 to Line 129 and Line 531 to Line 543). The effect of the loading rate on the deformation and fracture behavior is often ignored, which is usually acceptable since most of the rock engineering application can be regards as a static or quasi-static problem and the loading rate won`t vary in wide range. However, for the rock engineering application involving in strong dynamic loadings, such as rock blasting, it is essential to take into account the effect of loading rate on the dynamic deformation and fracture behavior of rock.

Finally, since the aim of this article is to research the pure model-II fracture patterns under static and dynamic loading conditions, and many novities are described in our previous research, the novities are only mentioned in the introduction section and explained in the methodology section, rather than comparing with the previous research.

The new result found in this article is that the crack initiates from the tip of the pre-fabricated notch only when the loading rates at lower values, such as 1m/s and 5m/s. While the loading rates continuously increase and reach higher values such as 10m/s or 50m/s, the failure patterns of rock are not influenced by the pre-fabricated notch. Therefore, it is obvious that the various loading rates would impact the rock failure behavior.

  1. In addition to the numerical modeling, what’s the experimental evidence of the crack initiation and progression at a high constant loading rate?

Authors response: Thanks for the comments and constructive suggestions.

We have compared the FDEM obtained results with numerical results conducted by Liu, Kou et al. (2007) [9] using R-T2D and experimental result carried out by Rao (1999) [10]. It is founded that the modelling results obtained by the FDEM in this research (Figure 6) agree well with that modelled by the R-T2D. (Figure 15), and the FDEM obtained fracture pattern is similar to that observed by Rao (1999) and Liu, Kou et al. (2007).

For more details it can be found in newly added Section 4.2 and Figure 15 (From Line 592 to Line 611).

  1. Page 7, Table 1, where do the tensile strength 20 MPa and compressive strength 200 MPa come from? So, the strength values are independent of the loading rate in the modeling part? In addition, how do the authors know the surface friction coefficient of 0.1?

Authors response: Thanks for the comments and constructive suggestions.

In terms of most of the geomaterials, such rock or concrete, the compress strength is 8-12 times greater than the tensile strength. So, the compressive strength of the rectangular beam used in this manuscript is assumed 200 MPa that is 10 times greater than the tensile strength. The various strength values of the rock material listed in Table 1 were general parameters of granite under relatively ideal condition.

  1. Section 4, what kind of simulations did the authors use to study the effect of loading rate on the rock behaviors? Compression, tension, or bending? Pls give an image of the model.

Authors response: Thanks for the comments and constructive suggestions.

The modelling results of Section 4 were obtained from the Brazilian tensile strength (BTS) test which has been mentioned in fifth paragraph (From Line 551 to Line 579). Due to there are 9 groups of BTS test be modelled in total, it would occupy too much space if all the simulation results be put in the manuscript. Therefore, only 4 groups of representative results are added in the Section 4.

  1. Page 15, Line 472, The FDEM obtained pure mode-II fracture toughness under the loading rate of 1m/s and 5m/s are and, respectively. What are the fracture toughness values at 10 m/s and 50 m/s?

Authors response: Thanks for the comments and constructive suggestions.

When the loading rate reaches 10 m/s or 50 m/s, the failure patterns of rock are different from those under lower loading rates. On these conditions, there is no stress concentration occurring in the tip of the pre-fabricated notch and the stress equilibrium cannot be achieved. Therefore, it makes no sense to calculate the pure mode-II fracture toughness under the loading rate of 10m/s and 50m/s.

  1. Page 15, equation 16, the compressive strength is used. How do the authors know that tensile strength and shear strength follow the same equation? By the way, how to evaluate the shear strength of rock? How does the shear strength affect the current numerical simulation of dynamic shear fracture of rocks?

Authors response: Thanks for the comments and constructive suggestions.

The correlation between the tensile strength and the loading rate can be assumed following more or less the Equation 16 similar to that for uniaxial compressive strength. This conclusion can be found in this research[11]. However, in the FDEM, the fracture process of rock is controlled by the fracture energy release rate rather than fracture toughness. The fracture energy release rate can be defined by the Equation (16). Therefore, the proposed method can model the rock failure under different loading rates without considering the rock strength individually.

The aim of this article is to research the effect of various loading rates on rock pure mode-II fracture by using the FDEM, rather than using the proposed method to evaluate the shear strength of rock, so this problem is not considered there.  Although the shear strength of rock is the key factor to affect the shear failure, the key potin of this article is to investigate the effect of different loading rates on shear fracture. Therefore, the shear strength of rock in this article is a fixed value whose influence on rock failure is not discussed there.

Reference

  1. An; Liu; Han, Hybrid finite–discrete element modelling of rock fracture process in intact and notched Brazilian disc tests. Eur. J. Environ. Civ. Eng. 2021, 1-34.
  2. Huaming; Hongyuan; han, Hybrid finite-discrete element modelling of rock fracture during conventional compressive and tensile strength tests under quasi-static and dynamic loading conditions. Latin American Journal of Solids and Structures 2020, 17 (6), 1-32.
  3. An; Liu; Han, Hybrid Finite-Discrete Element Modelling of Excavation Damaged Zone Formation Process Induced by Blasts in a Deep Tunnel. Advances in Civil Engineering 2020, 2020, 7153958.
  4. An; Fan; Liu et al., The State of the Art and New Insight into Combined Finite-Discrete Element Modelling of the Entire Rock Slope Failure Process. Sustainability 2022, 14 (9), 4896.
  5. Liu; Kou; Lindqvist et al., Numerical studies on the failure process and associated microseismicity in rock under triaxial compression. Tectonophysics 2004, 384 (1), 149-174.
  6. Liu, A numerical model for failure and collapse analysis of geostructures. Australian Geomechanics 2010, 45 (3), 11-19.
  7. Munjiza, The Combined Finite-Discrete Element Method. Wiley Online Library: 2004.
  8. Xiang; Munjiza; Latham, Finite strain, finite rotation quadratic tetrahedral element for the combined finite–discrete element method. International journal for numerical methods in engineering 2009, 79 (8), 946-978.
  9. Liu; Kou; Lindqvist et al., Numerical modelling of the heterogeneous rock fracture process using various test techniques. Rock mechanics and rock engineering 2007, 40 (2), 107-144.
  10. Rao. Pure shear fracture of brittle rock: a theoretical and laboratory study. Luleå tekniska universitet, 1999.
  11. Zhao, Applicability of Mohr–Coulomb and Hoek–Brown strength criteria to the dynamic strength of brittle rock. International Journal of Rock Mechanics and Mining Sciences 2000, 37 (7), 1115-1121.

Reviewer 2 Report

1. The authors should clearly explain in the introduction what the novelty of this research is compared to their previous study (An et al. 2022)?

Section 3.1 (Figs. 6 and 7) and Section 4 (Fig. 12) were previously presented by the authors in the paper by An et al. (2022). The only new sections in this manuscript are Sections 3.2 and 3.3. It is better to mention these results in the introduction section and present the new results in the results section. Therefore, the overall structure of the paper needs to be fundamentally revised.

An H, Wu S, Liu H, Wang X. 2022. Hybrid Finite-Discrete Element Modelling of Various Rock Fracture Modes during Three Conventional Bending Tests. Sustainability. 14(2): 592.

2. The authors mentioned that the ability and reliability of the FDEM in modelling rock fracture should be confirmed first and therefore referred to previous research (Lines 259-274). This can only be a reason to use this method for simulation. Validation of the numerical model should be done independently in this research. Before the numerical simulation is used, it has to be adequately validated to real case histories or experimental data. The authors can use the experiments conducted by other researchers.

3. According to the manuscript, the paper's title should be modified and it is better to refer to the four-point bending test in the title.

4. On what basis were the parameters of Table 1 selected? If they are obtained from laboratory measurements, it should be explained in the article. The unit of internal friction angle is degree (°), not °C. In addition, the coefficient of internal friction (μ) does not have any unit.

5. The manuscript contains many grammatical and language errors that need to be corrected to meet the level of the journal. For example:

·       Line 24: “fracture patters”: fracture patterns

·       Line 24: “On these conditions”: Under these conditions

·       Line 26: “curves shows”: curves show

·       Line 45: “to obtained”: to obtain

·       Line 85: “that supposed as”: that is supposed to

·       Line 95: “a few number of multiscale”: a few multiscale

·       Line 118: “was initially develop”: was initially developed

·       Line 323: “its ability of carrying”: its ability to carry

Author Response

Dear Reviewer

Thanks very much for reviewing our manuscript and giving us constructive suggestion and comments. We have revised our manuscript according to your review and comments.

The following is a summary of changes against each point which is being raised by you. In the following, the paper submitted for review in last time is referred here as the original manuscript and this time is referred as revised manuscript. Two copies of the revised manuscripts are to be submitted in this time in one PDF file, the first part of the PDF file is the marked copy, where the changes are marked using the function of MS Word`s “Track Changes”, while the second part is the normal copy of the main manuscript. It is worth noting that the number of lines mentioned below is based on the marked copy.

In the following, we firstly list the review comments/suggestions which are raised by your then present our responses. When presenting our responses, we firstly indicate whether we agree/disagree. We then explain how the changes are made to incorporate your review comments and where are the changes in the marked copy of the revised manuscript if we agree your comments. Otherwise, the reason is to be explained.

Specific comments:

  1. The authors should clearly explain in the introduction what the novelty of this research is compared to their previous study (An et al. 2022)?

Authors’ response: Thanks for the comments and constructive suggestions.

We agree the comment and think it is a good idea to improve our manuscript. We have added those explains at the last paragraph of Introduction Section.

The FDEM has been proposed by the authors to model the conventional rock mechanical tests, e.g. Brazilian dis test[1], Uniaxial compressive test[2], and implemented in many engineering problems, e.g. tunnel excavation[3], slope failure analysis[4]. The previous studies focus on the macro phenome of the rock mechanism, while this research try to illustrate the micro phenomenon of the rock fracture process. This paper aims at demonstrating the capabilities of the FDEM in modelling the dynamic behavior throughout the course of pure mode-II fracture and the combined method's capacity to accurately catch how loading rate affects dynamic fracture toughness. To successfully model the pure mode-II fracture process, the transition from continuum to discontinuum, the three fracture modes and the effect of the loading rate are considered.

 The novelties of the FDEM are described in detail as follows. 

Firstly, a hybrid finite-discrete element method is proposed and developed on the basis of the previous achievement in literature to overcome the limitations of the continuum method (e.g. finite element method) or discontinuum method (e.g. discrete element method) in modelling the complete rock fracture and fragmentation process. The hybrid finite-discrete element method takes the advantages of the both the continuum methods and discontinuum methods and can model the transition from continuum to discontinuum. Then, an integrated development environment is development for the hybrid finite-discrete element method, i.e. Y-HFDEM. The Y-HFDEM is development using C++ and OpenGL on the basis of the author`s 2D and 3D on the basis of their previous enriched finite element codes RFPA-RT2D [5] and TunGeo3D [6], and the open-source combined finite-discrete element libraries Y2D and Y3D originally developed by Munjiza (2004)[7] and Xiang et al. (2009) [8], respectively.

Secondly, the authors proposed three fracture models for model the Mode-I (tensile failure), Mode-II (shear failure) and mixed mode I-II fractures. The combined single and smeared crack model was developed by Munjiza et al. (1999) for modelling tensile failure (Mode-I) of concrete only in the Y2D/ 3D open-source. The Y-HFDEM extends the combined single and smeared crack model for modeling various fracture models by proposing three fracture models. Thus, the hybrid finite-discrete element method not can only model tensile failures but also can model shear failures and mixed fractures.

Thirdly, the Y-HFDEM takes the effect of loading rate into account by implementing an empirical relation between the static strengths and the dynamic strengths derived from the dynamic rock fracture experiments (From Line 127 to Line 129 and Line 531 to Line 543). The effect of the loading rate on the deformation and fracture behavior is often ignored, which is usually acceptable since most of the rock engineering application can be regards as a static or quasi-static problem and the loading rate won`t vary in wide range. However, for the rock engineering application involving in strong dynamic loadings, such as rock blasting, it is essential to take into account the effect of loading rate on the dynamic deformation and fracture behavior of rock.

Finally, since the aim of this article is to research the pure model-II fracture patterns under static and dynamic loading conditions, and many novities are described in our previous research, the novities are only mentioned in the introduction section and explained in the methodology section, rather than comparing with the previous research.   

  1. Section 3.1 (Figs. 6 and 7) and Section 4 (Fig.12) were previously presented by the authors in the paper by An et al. (2022). The only new sections in this manuscript are Sections 3.2 and 3.3. It is better to mention these results in the introduction section and present the new results in the results section. Therefore, the overall structure of the paper needs to be fundamentally revised.

Authors’ response: Thanks for the comments and constructive suggestions.

Compared with previous research (An et al. 2022), the Section 3.1 of this manuscript descripted the stress distribution in detail besides illustrated the cracks initiation and propagation in the 4PB test under loading rate of 1m/s. Although there are some repetitions to a certain degree, the Section 3.1 is the indispensable part in this article to study the effect of loading rates on rock failure behavior. In order to reflect the tendency of results obtained by the FDEM more accurately and exactly, more simulations of Brazilian tensile strength (BTS) test under different loading rates were conducted in the Section 4 of this article (From Line 551 to Line 579). The result of previous study has been mentioned in Introduction Section (From Line 122 to Line 127) and the new results has been recorded in the Conclusion Section (From Line 631 to Line 642).

  1. The authors mentioned that the ability and reliability of the FDEM in modelling rock fracture should be confirmed first and therefore referred to previous research (Lines 259-274). This can only be a reason to use this method for simulation. Validation of the numerical model should be done independently in this research. Before the numerical simulation is used, it has to be adequately validated to real case histories or experimental data. The authors can use the experiments conducted by other researchers.

Authors’ response: Thanks for the comments and constructive suggestions.

We have calibrated the hybrid finite-discrete method in modelling the rock fracture using the Brazilian disc test in the Section 3 (From Line 315 to Line 335). The FDEM modelled results are compared with the experimental result and the typical rock failure process of BTS test in literatures. The modelled result agrees well with the experimental result. In addition, the input tensile strength for the FDEM mode is compared with the FDEM obtained results, which is 1.18 time larger than the input value due to the effect of the loading rate.

As we have done the calibration work in our previous published papers and we do not focus on the validation work of the FDEM in this research, we do not offer more calibration work here, instead, the references are provided.  

  1. According to the manuscript, the paper's title should be modified and it is better to refer to the four-point bending test in the title.

Authors’ response: Thanks for the comments and constructive suggestions.

The keywords of “four-potin bending test” have been added in the paper's title.

  1. On what basis were the parameters of Table 1 selected? If they are obtained from laboratory measurements, it should be explained in the article. The unit of internal friction angle is degree (°), not °C. In addition, the coefficient of internal friction (μ) does not have any unit.

Authors’ response: Thanks for the comments and constructive suggestions.

The rock parameters listed in the Table 1 are identified by selecting the general parameters of granite. It is a relatively ideal condition for the modelling. The problems about units have been corrected in Table 1.

  1. The manuscript contains many grammatical and language errors that need to be corrected to meet the level of the journal. For example:
  • Line 24: “fracture patters”: fracture patterns
  • Line 24: “On these conditions”: Under these conditions
  • Line 26: “curves shows”: curves show
  • Line 45: “to obtained”: to obtain
  • Line 85: “that supposed as”: that is supposed to
  • Line 95: “a few number of multiscale”: a few multiscale
  • Line 118: “was initially develop”: was initially developed
  • Line 323: “its ability of carrying”: its ability to carry

Authors’ response: Thanks for the comments and constructive suggestions.

All the grammatical and language errors have been corrected in this manuscript.

Reference

  1. An; Liu; Han, Hybrid finite–discrete element modelling of rock fracture process in intact and notched Brazilian disc tests. Eur. J. Environ. Civ. Eng. 2021, 1-34.
  2. Huaming; Hongyuan; han, Hybrid finite-discrete element modelling of rock fracture during conventional compressive and tensile strength tests under quasi-static and dynamic loading conditions. Latin American Journal of Solids and Structures 2020, 17 (6), 1-32.
  3. An; Liu; Han, Hybrid Finite-Discrete Element Modelling of Excavation Damaged Zone Formation Process Induced by Blasts in a Deep Tunnel. Advances in Civil Engineering 2020, 2020, 7153958.
  4. An; Fan; Liu et al., The State of the Art and New Insight into Combined Finite-Discrete Element Modelling of the Entire Rock Slope Failure Process. Sustainability 2022, 14 (9), 4896.
  5. Liu; Kou; Lindqvist et al., Numerical studies on the failure process and associated microseismicity in rock under triaxial compression. Tectonophysics 2004, 384 (1), 149-174.
  6. Liu, A numerical model for failure and collapse analysis of geostructures. Australian Geomechanics 2010, 45 (3), 11-19.
  7. Munjiza, The Combined Finite-Discrete Element Method. Wiley Online Library: 2004.
  8. Xiang; Munjiza; Latham, Finite strain, finite rotation quadratic tetrahedral element for the combined finite–discrete element method. International journal for numerical methods in engineering 2009, 79 (8), 946-978.

Reviewer 3 Report

This manuscript investigates the rock fracturing process by using hybrid finite-discrete element method that is interesting, and can be published after necessary corrections as follows:

1- Please discuss more in the introduction about the relevant published published works to provide more evidence of the innovation of your work.

2- Please describe the methodology, step by step, as applied in the text body, at the end of introduction. 

3- The quality of some figures, for instance Figs. 6, 8, 10 is not good, please improve them. 

4- Please present further discussion of the results and compare which with other relevant results in the literature.

5- Please compare the results of numerical simulation with the experimental results to calibrate the parameters of the proposed model. 

Author Response

Dear Reviewer

Thanks very much for reviewing our manuscript and giving us constructive suggestion and comments. We have revised our manuscript according to your review and comments.

The following is a summary of changes against each point which is being raised by you. In the following, the paper submitted for review in last time is referred here as the original manuscript and this time is referred as revised manuscript. Two copies of the revised manuscripts are to be submitted in this time in one PDF file, the first part of the PDF file is the marked copy, where the changes are marked using the function of MS Word`s “Track Changes”, while the second part is the normal copy of the main manuscript. It is worth noting that the number of lines mentioned below is based on the marked copy.

In the following, we firstly list the review comments/suggestions which are raised by your then present our responses. When presenting our responses, we firstly indicate whether we agree/disagree. We then explain how the changes are made to incorporate your review comments and where are the changes in the marked copy of the revised manuscript if we agree your comments. Otherwise, the reason is to be explained.

Comments made by Reviewer

This manuscript investigates the rock fracturing process by using hybrid finite-discrete element method that is interesting, and can be published after necessary corrections as follows:

Authors’ response: Thank you very much for your positive comments.

Specific comments:

1. Please discuss more in the introduction about the relevant published works to provide more evidence of the innovation of your work.

Authors’ response: Thanks for the comments and constructive suggestions.

We have discussed the relevant published works in the last paragraph of the Introduction Section (From Line 104 to Line 114 and Line 146 to Line 150) which are marked using blue color. The innovations of the work can be summarized as follows.

Firstly, it combines the finite element method (FEM) with the discrete element method (DEM) and inherits the advantages of FEM in describing elastic deformations, and the capabilities of DEM in capturing interactions and fracturing processes of solids [1, 2].

Secondly, the hybrid finite-discrete element method (FDEM) was initially proposed to model the pure mode-I fracture for the concrete only by Munjiza[2], while the FDEM in this research has extended to model the three fracture models, i.e. pure mode-I, pure mode-II and mixed mode I-II. (More details can be found in the Methodology Section)

Last but not least, to truly reflect the effect of various loading rates on rock failure behavior, an empirical correlation between the static and the dynamic strengths is implemented in the FDEM. Thus the FDEM can well capture the dynamic behavior of rock during the impact loading (From Line 127 to Line 129 and Line 531 to Line 543).

2. Please describe the methodology, step by step, as applied in the text body, at the end of introduction. 

Authors’ response: Thanks for the comments and constructive suggestions.

We have described the organization of the text body as follows. Section 2 presents the core concepts of the hybrid finite-discrete element method, especially introduces the transition from continuum to discontinuum through rock fracture, which is considered as the key component. Section 3 describe the numerical models for the four-point bending test, then various loading rate are applied on the rock samples. The pure mode-II fracture process are analyzed and the fracture toughness are obtained in this section. In Section 4, the effect of the loading rate is discussed. Finally, the conclusions from this study are drawn.

In addition, we have given the modelling process step by step as follows. In the process of modelling the 4PB test, a numerical model need be established, and all the rock parameters should be set up in advanced. Next, different loading rates should be set on the two upper loading rolls. Through calculating the energy release rate of mode II fracture which can be defined by the empirical correlation, the sliding displacement of crack mouth can be determined and further the transition from continuum to discontinuum achieves. Finally, the rock sample’s pure mode-II fracture patterns under various loading rates can be illustrated visually. It is worth noting that the contact detection implanted in the FDEM is employed to reflect the interaction between the elements in order to save the modelling time.

For more details can be found at the end of Introduction Section (From Line 130 to Line 145).

3. The quality of some figures, for instance Figs. 6, 8, 10 is not good, please improve them. 

Authors’ response: Thanks for the comments.

We have tried our best to improve the qualities of the mentioned Figures (See Figure 6, 8 and 10). However, as the FDEM is developed by the research team instead of a commercial company, the qualities of the output figures are still need a long time to improve.

4. Please present further discussion of the results and compare which with other relevant results in the literature.

Authors’ response: Thanks for the comments.

We have compared the FDEM obtained results with numerical results conducted by Liu, Kou et al. (2007) [3] using R-T2D and experimental result carried out by Rao (1999) [4]. It is founded that the modelling results obtained by the FDEM in this research (Figure 6) agree well with that modelled by the R-T2D. (Figure 14), and the FDEM obtained fracture pattern is similar to that observed by Rao (1999) and Liu, Kou et al. (2007).

For more details it can be found in newly added Section 4.2 and Figure 15 (From Line 592 to Line 611).

5. Please compare the results of numerical simulation with the experimental results to calibrate the parameters of the proposed model. 

Author`s response: Thanks very much for your comments.

We agree that it is very important to validate the numerical code parameters. Frankly, it is tough work and we might not be able to deliberate this in one part of an academic paper. Thanks very much for the comments and we are planning to do this for a paper to calibrate the numerical code parameters.

For this research, as we have done a general calibration of the numerical model by the comparison with the Brazilian disc test[5] and the uniaxial compressive test[6], we just cited them directly.  

Reference

  1. Zhao, Q., et al., Numerical simulation of hydraulic fracturing and associated microseismicity using finite-discrete element method. Journal of Rock Mechanics and Geotechnical Engineering, 2014. 6(6): p. 574-581.
  2. Munjiza, A., The Combined Finite-Discrete Element Method. 2004: Wiley Online Library.
  3. Liu, H., et al., Numerical modelling of the heterogeneous rock fracture process using various test techniques. Rock mechanics and rock engineering, 2007. 40(2): p. 107-144.
  4. Rao, Q., Pure shear fracture of brittle rock: a theoretical and laboratory study. 1999, Luleå tekniska universitet.
  5. An, H., H. Liu, and H. Han, Hybrid finite–discrete element modelling of rock fracture process in intact and notched Brazilian disc tests. European Journal of Environmental and Civil Engineering, 2021: p. 1-34.
  6. Huaming, A., L. Hongyuan, and H. han, Hybrid finite-discrete element modelling of rock fracture during conventional compressive and tensile strength tests under quasi-static and dynamic loading conditions. Latin American Journal of Solids and Structures, 2020. 17(6): p. 1-32.

Round 2

Reviewer 1 Report

The authors address the comments well. The revised manuscript is improved and is recommended for publication.

Reviewer 2 Report

The authors responded to all the comments from my previous review.

Reviewer 3 Report

The raised comments were well addressed. I have no more comment.